# Mathematical model linking telomeres to senescence in *Saccharomyces cerevisiae* reveals cell lineage versus population dynamics

Anaïs Rat [1,2,6], Veronica Martinez Fernandez[3], Marie Doumic[2,4,7] ✉, Maria Teresa Teixeira [3,7] ✉ & Zhou Xu [5,7]

Telomere shortening ultimately causes replicative senescence. However, identifying the mechanisms driving replicative senescence in cell populations is challenging due to the heterogeneity of telomere lengths and the asynchrony of senescence onset. Here, we present a mathematical model of telomere shortening and replicative senescence in *Saccharomyces cerevisiae* which is quantitatively calibrated and validated using data of telomerase-deficient single cells. Simulations of yeast populations, where cells with varying proliferation capacities compete against each other, show that the distribution of telomere lengths of the initial population shapes population growth, especially through the distribution of cells' shortest telomere lengths. We also quantified how factors influencing cell viability independently of telomeres can impact senescence rates. Overall, we demonstrate a temporal evolution in the composition of senescent cell populations—from a state directly linked to critically short telomeres to a state where senescence onset becomes stochastic. This population structure may promote genome instability and facilitate senescence escape.

Mathematical modeling as an approach plays a pivotal role in elucidating the underlying principles and mechanisms governing biological processes. However, it is challenging to develop models that capture the heterogeneity of biological systems particularly when heterogeneity evolves over time during population growth. This occurs during tumorigenesis, for example, where increasing genomic instability can affect cell morphology, proliferation rates, invasion potential and resistance mechanisms[1]. Another example is the phenotypic heterogeneity that accumulates over time in the course of replicative senescence caused by telomere shortening[2].

In human somatic cells, telomere shortening eventually activates the DNA damage checkpoint, arrests the cell cycle and triggers replicative senescence, where cells remain metabolically viable, but no longer divide[3]. Replicative senescence has thus been implicated in cellular aging and in the regenerative capacity of tissues. The loss of telomere length homeostasis is also a characteristic of cancer cells, which escape replicative senescence early in the transformation process[4]. These rare events can happen through the reactivation of telomerase expression or through telomere alternative lengthening pathways, with break-induced replication being a primary mechanism,

[1]Aix Marseille Univ, CNRS, I2M, Centrale Marseille, Marseille, France. [2]Sorbonne Université, CNRS, Université de Paris, Inria, Laboratoire Jacques-Louis Lions UMR7598, Paris, France. [3]Sorbonne Université, CNRS, Laboratoire de Biologie Moléculaire et Cellulaire des Eucaryotes, LBMCE, Paris, France. [4]CMAP, Inria, IP Paris, Ecole polytechnique, CNRS, Palaiseau, France. [5]Sorbonne Université, CNRS, Laboratory of Computational and Quantitative Biology, LCQB, Paris, France. [6]Present address: Univ Brest, CNRS UMR 6205, Laboratoire de Mathématiques de Bretagne Atlantique, Brest, France. [7]These authors jointly supervised this work: Marie Doumic, Maria Teresa Teixeira, Zhou Xu. ✉e-mail: Marie.Doumic@inria.fr; Teresa.Teixeira@cnrs.fr

as first described in budding yeast[5-7]. Individuals displaying abnormally short telomeres are susceptible to a spectrum of deadly telomeropathies, which are currently incurable[8]. Therefore, telomere length has emerged as a biological marker of aging and various diseases[9,10]. However, the mechanisms linking telomere length and shortening with replicative senescence at a cell population level remain poorly understood[2,11-15].

Deciphering phenotypic heterogeneity in replicative senescence is challenging in cell populations because senescence often appears as a homogenous and progressive process. In contrast, in single-cell analyses, senescence displays considerable cell-to-cell variability[2,16-18]. For instance, in telomerase-negative *Saccharomyces cerevisiae* single cells, in which consecutive cell divisions were monitored for multiple generations, the onset of senescence varied greatly. This variability was observed even among the progeny of the same cell, making it inherently asynchronous. Senescence onset was also found to be often abrupt: cells switch in a single division from fast proliferation to a prolonged cell cycle, which can be followed by a few more long cell cycles before cell death[17,18]. Below we use the term replicative senescence or senescence to refer to this ultimate sequence of long cell cycles, qualified as terminal since they lead to cell death in the budding yeast model.

Numerical and experimental evidence in *S. cerevisiae* supports a model in which the onset of senescence is triggered by the shortest telomere reaching a threshold length below which the DNA damage checkpoint is first activated[15,19-22]. Alternative hypotheses proposing that replicative senescence could be triggered by multiple telomeres or simply by the passage of time fail to explain the substantial variability observed in the onset of senescence. Therefore, a significant contributor to asynchrony arises from the variable shortening trajectory of the shortest telomeres within a cell. This trajectory, called *type A*, is driven by the process of telomere replication, which inherently generates asymmetry and variability, as observed at the molecular level in *S. cerevisiae* and other organisms[23-26].

The experiments in single telomerase-negative budding yeast cells also revealed a second source of heterogeneity[17,18]. Specifically, a subset of individual cells, called *type B* cells, undergo several switches between non-terminal cell cycle arrests and normal proliferation prior to terminal senescence, in contrast to *type A* cells, which undergo the switch in a single step (Fig. 1a-c, Supplementary Fig. 1a). Even though both terminal and non-terminal arrests are abnormally long cycles associated with the activation of the DNA damage checkpoint, genetic and statistical analyses showed that their manifestation corresponds to distinct genetic pathways and probability laws, thus possibly arising from distinct molecular origins[18,27,28]. In non-terminal arrests of *type B* cells, the cells either recover upon a Pol32-dependent repair of damage, presumably stemming from a yet unidentified telomere defect or adapt to the damage[27], forcing mitosis until successful repair. These cycles of mitosis in the presence of unrepaired DNA damage are a source of genomic instability.

These detailed observations at the single-cell level have thus provided important insights into individual cell trajectories leading to senescence. However, for understanding tissue development, aging, cancer development, and growth, the biologically relevant scale is not lineages, but rather the population. Within a population, complex effects of competition and selection emerge, which cannot be fully explored through lineage studies alone. Moreover, telomere shortening, cell death, and terminal and non-terminal arrests exhibit intricate interactions in overall population dynamics, distinct from their effects on lineage observations. The challenge lies in the fact that in population experiments, we can only measure averages at single points in time. For instance, parameters such as generational age or the length of the shortest telomeres triggering replicative senescence cannot currently be directly measured experimentally in populations.

In this study, using *S. cerevisiae* as a model organism, we seek to elucidate the temporal dynamics and structural organization of

telomerase-negative cell populations, focusing on how key cellular and telomeric parameters evolve within this complex system. We present a mathematical model of consecutive cell divisions limited by telomere erosion and simulated cell populations from telomerase inactivation to senescence. Our current model is based on extensive knowledge in *S. cerevisiae*, in which the mechanism of telomere shortening and the molecular pathways underlying replicative senescence are well described[2,24,29-32]. We incorporate in our new model assumptions derived from our previous statistical analyses of *type A* and *type B* lineages: *type A* cells enter senescence when their shortest telomere reaches a specific, deterministic length[20]. Prior to this, they may enter a (first) non-terminal arrest, and become *type B*, with a probability that increases as their shortest telomere shortens[28]. We also formulate a probability law to describe senescence entry in *type B* cells, which has not been done in previous studies. The parameters of our model are then further refined by calibration on quantitative data obtained in microfluidics experiments of telomerase-negative single-cell lineages[18,33].

Our results show that, in budding yeast, the early stage of a telomerase-negative population is driven by an almost deterministic limit of telomere length that triggers senescence. This stage enables the rapid selection of cells from the initial pool that display the longest and shortest telomeres. Therefore, the size and the structure of the initial population prior to telomerase-inactivation is key to determine the proliferative capacity of the progeny. In contrast, the late-stage population consists of cells undergoing a stochastic route to senescence, mostly no longer dependent on initial telomere lengths. We speculate that this late-stage population offers potential pathways for senescence escape. Additionally, our results enable an assessment of the impact of other independent comorbidities on lifespan, possibly enhancing our understanding of age-related human diseases.

## Results
### Rationale of our integrative approach
Our aim was to build a population model and use in silico experiments to investigate the structure and evolution of a population of yeast cells in which telomerase is inactivated. The strategy is to mimic actual senescence experiments in which population doublings are monitored each day from telomerase inactivation to senescence crisis, i.e. when the culture reaches the lowest growth capacity, typically a few days after telomerase inactivation. Because cells grow very robustly and exhaust nutrients during the first days of these experiments, cultures are diluted each day in fresh media, so only a fraction of the population is sampled for the next day. In this assay, the cell concentration and the statistical mode of telomere lengths, i.e. the peak of the telomere length distribution, can be measured each day (Fig. 1a).

This conventional population assay introduces several biases. First, cell-to-cell variability in cell-cycle durations introduces competition (fast-dividing cells quickly take over the population). Second, senescence-induced death introduces selection effects: fast-dividing cells, although initially over-represented, are likely to die first because they reach their proliferative limit more quickly. In contrast, slow-dividing cells can take over the population in the long term. Therefore, in a replicative senescence experiment, the number of population doublings is not linearly related to the number of generations undergone by cells. For this reason, daily telomere length measurements cannot be used directly to estimate a shortening rate per generation. In addition, daily dilution introduces a sampling bias. All these complex mechanisms require the use of a comprehensive model to analyze their intricate effects.

To build the mathematical model, we used data from single cell lineages observed in microfluidics experiments, corresponding to more than 5000 individual cell divisions[18,33], bringing together all our previous analyses[20,22,28] in a comprehensive study. In microfluidics experiments, the consecutive cell cycle durations are measured from

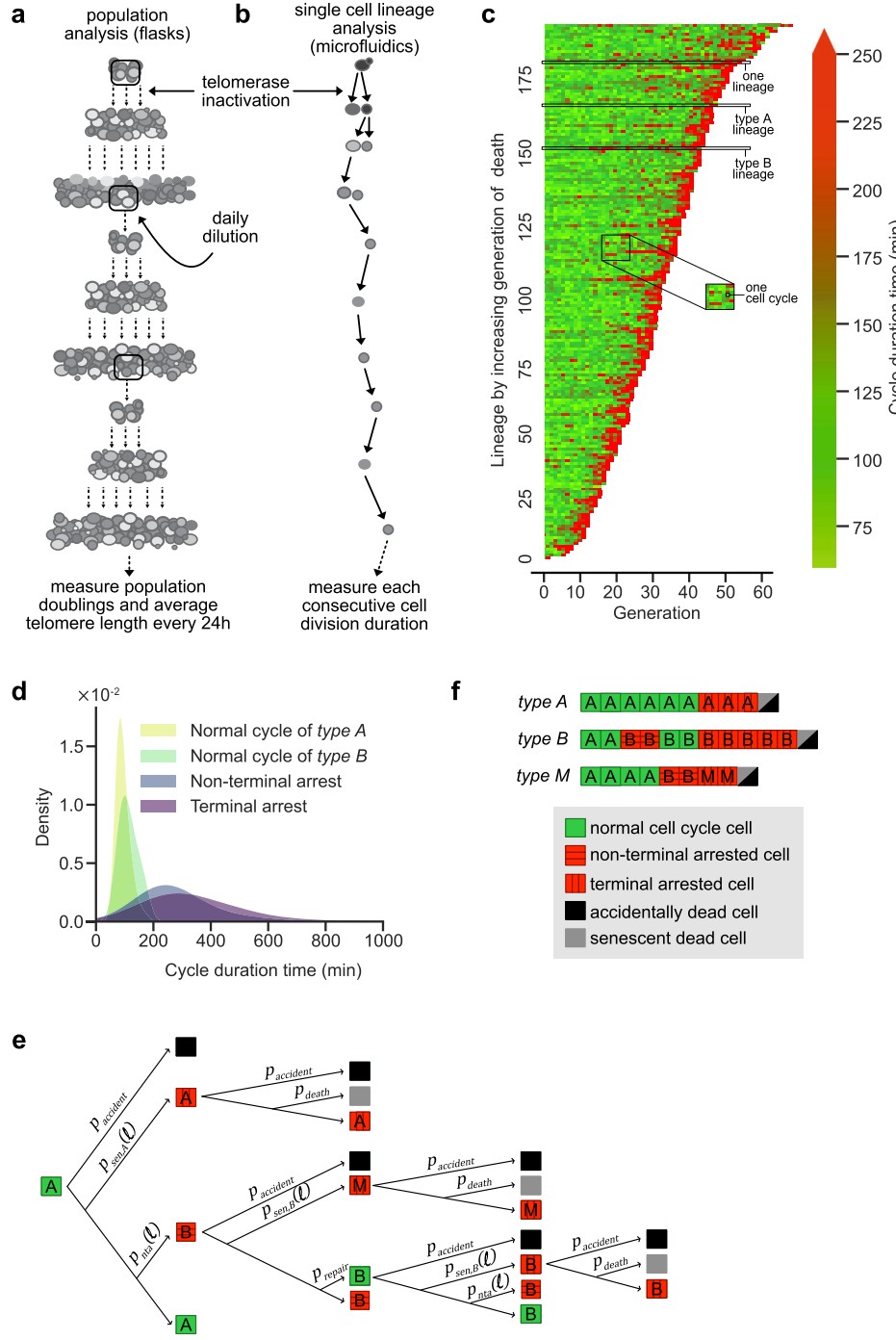

**Fig. 1 | Principle of the mathematical model of replicative senescence.** Schematic representation of population (**a**) and single-cell lineage (**b**) experiments. **c** Experimental data of microfluidics experiments from[27]. Display of consecutive cell cycle durations of lineages of a yeast strain harboring a subunit of telomerase under the control of a conditionally repressible promoter. Generation 0 corresponds to the moment from which telomerase is inactivated. Each horizontal line is an individual cell lineage, and each segment is a cell cycle. Cell cycle duration (in minutes) is indicated by the color bar on the right. Examples of type A or type B lineages are shown. **d** Kernel density smoothing representation of the distributions of the cell cycle durations for indicated cell types extracted from experimental data in (**c**) (see Supplementary Methods Fig. 1a). **e** Tree diagram of the mathematical model. It indicates the several fates of each cell type and respective probability rates, some being functions of the length of the shortest telomere in the cell $\ell$. The colors of the squares indicate whether the cell cycle is normal (green) or abnormally long (red). **f** Three examples of lineages containing indicated type cells (A, B, or M). The colors of the squares are as in (**e**). Vertical or horizontal strips specify whether the cell cycles are terminal (never followed by a normal cell cycle) or not, respectively. Black or gray cells indicate the type of death.

telomerase inactivation to cell death in individual cell lineages (Fig. 1b, c). These quantitative data evidenced two main senescence pathways, *type B* or *type A*, with or without non-terminal arrest, respectively, and enabled us to estimate their relative frequency. Non-terminal arrests and cycle distributions have been extracted from these quantitative data based on a threshold D = 180 min, whose value

and robustness were discussed in ref. [28], allowing us to differentiate long cycles from normal cycles (Fig. 1d).

The model involves a detailed description of the population at the single cell level, entirely based on experimental observations (Fig. 1c, d): each cell is tracked in time through its state $S = (L, T, C, \tau, X)$, in which:

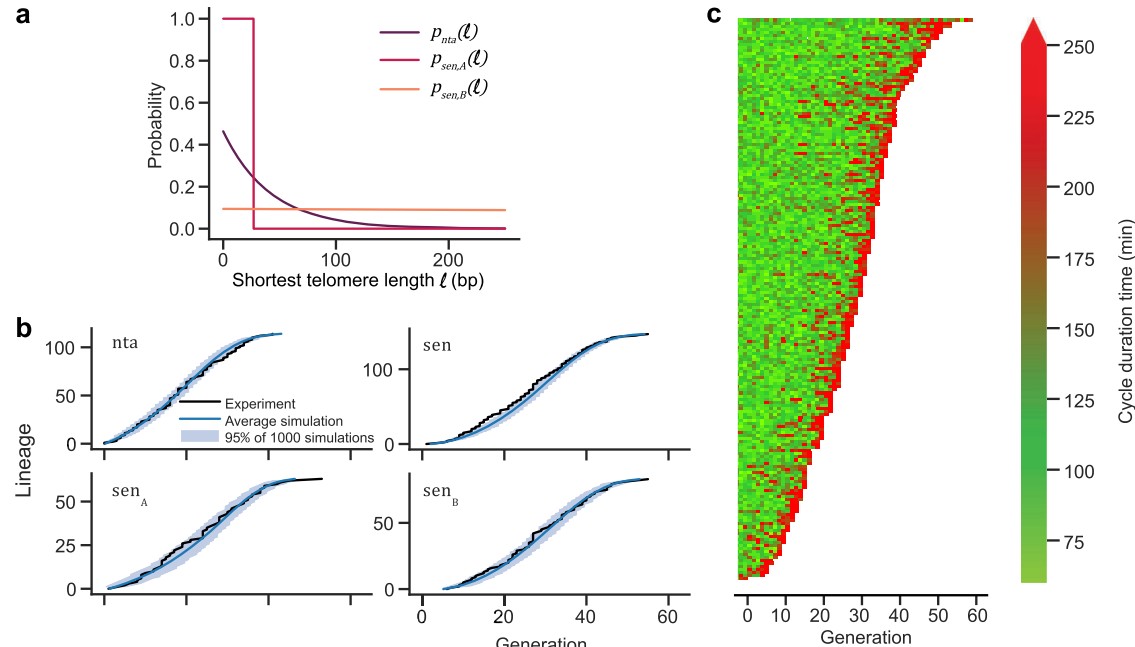

**Fig. 2 | Mathematical model of telomerase-deficient single lineages. a** Best fits of indicated probability rates as functions of the length of the shortest telomere $\ell$. **b** Comparison between 1000 random simulations of the best-fit model of single lineages and experimental data expressed as generations of arrest classified by lineage type and type of arrest, as extracted from Fig. 1c and Supplementary

Fig. 1b, c. nta: generation of the first non-terminal arrest; sen$_A$, sen$_B$: generation of first senescent arrest for type A or B cells; sen: generation of first senescent arrest for all types. Light blue area corresponds to 95% of 1000 simulations. **c** Simulated data of one microfluidics experiment with the best-fit mathematical model (parameters listed in Table 1). Other examples are shown in Supplementary Fig. 2a, b.

- $L \in Z^{2 \times 16}$ are the lengths of all telomeres in the cell (there are $16 \times 2 = 32$ telomeres in a haploid *S. cerevisiae* cell, which contains 16 chromosomes);
- $T \in \{A, B, M\}$ is the cell type (see below for a more detailed explanation of *type M*. In brief, *type M* is a subclass of *type B* cells, introduced in the model to account for *type B* lineages that cannot be experimentally distinguished from *type A*);
- $C \in \{nor, nta, sen\}$ is the cell cycle type (normal duration cell cycle (*nor*) or prolonged cell cycle. Prolonged cell cycle durations include non-terminal arrest (*nta*) or terminal (senescence) arrest (*sen*), respectively);
- $\tau \in R_+$ is the cell cycle duration time;
- $X$ is a set of individual variables of interest (generation since telomerase inactivation, birth time, ancestor index in the initial population, etc.).

Right after telomerase inactivation, we consider that all cells are initially *type A*, since *type A* is the normal type of cells in a telomerase-positive state, where abnormally-long cell cycles are very rarely observed. Similarly, we depart from a telomere length distribution originating from the telomerase-positive model described in refs. 15,20 (minor modifications on this distribution are described in the Methods section and Supplementary Fig. 1b).

The cells then evolve by consecutive cell divisions. At each cell division, the telomeres shorten according to the well-documented telomere shortening model described in ref. 22. The type and cycle type of the daughter then depend on its mother's state and on its shortest telomere length thanks to transition probabilities laws (Fig. 1e, f). Finally, the cell cycle duration of each cell is drawn from the experimental distribution corresponding to its type and cycle type (Fig. 1d).

In a previous statistical analysis[28], we found that the probability law governing the appearance of non-terminal arrest ($p_{nta}$) was distinct from the one describing senescence onset ($p_{sen,A/B}$) - for instance, the first non-terminal arrests in *type B* appear significantly earlier than the terminal arrests in *type A*. Also, the consecutive non-terminal arrests of

*type B* are suppressed in cells lacking functional Pol32, Cdc5, or Tid1/Rdh54, involved in DNA repair[18,27]. Yet, we must consider the scenario where a *type B* cell enters senescence during its first series of non-terminal arrests, in which case the experimental observation cannot distinguish it from a standard *type A*: this is why we introduce *type M*, which are *type B* cells experimentally misclassified as *type A*. *Type M* can only be known in simulations where we know the ground truth, and we find that their impact on the quantitative results is marginal.

**A deterministic threshold for type A, a fully stochastic fate for type B**

The laws of non-terminal and senescent arrests $p_{nta}$, $p_{sen,A}$ and $p_{sen,B}$ were fitted altogether on the basis of the generations of arrest from the experimental microfluidics data (Supplementary Fig. 1c, d), using the CMA-ES optimization algorithm (Fig. 2a and Table 1). We found that combining the two *type A* and *type B*[22,28] models into a comprehensive model resulted in remarkably good agreement with the data (Fig. 2b). We also found that the senescence law of *type A* ($p_{sen,A}$) is mostly deterministic with respect to the length of the shortest telomere in the cell, which is coherent with our previous findings[20]. However, in contrast, the best fit for $p_{sen,B}$ turned out to be completely independent of telomere length (Fig. 2a). This delineates two clearly distinct pathways to senescence. A cell can thus follow one of two scenarios:

- It divides for a number of generations of normal cycles until one telomere reaches the almost deterministic threshold $l_{min_A} = 27bp$. This is the canonical pathway of *type A* lineages.
- Or experiences at least one non-terminal arrest to become *type B* with probability $p_{nta}$. Then, the cell has a constant probability of entering senescence at each cell division (Fig. 2a). This results in greater variability in the length of the telomere triggering senescence of *type B* (possibly longer or much shorter than $l_{min_A}$). Therefore, senescence of *type B* lineages becomes independent of the length of the shortest telomere, and thus from the initial telomere length distribution.

With these laws calibrated, we next simulated the evolution of cell lineages as grown and analyzed in the microfluidics experimental setting (Fig. 2c, Supplementary 2a, b). The obtained profile was remarkably similar to the experimental one (Fig. 1c). By considering the simulated data for 1000 independent virtual experiments (Supplementary Fig. 2c), we were able to quantitatively compare the experimental data with the simulated one. The median and variance of the experimental lifespan of lineages fit very well within the confidence interval values obtained in the simulated data. Likewise, the simulated

percentage of *type B* lineages lies between 61% and 89%, consistent with the range of experimental *type B* proportion of 61-67% (see Supplementary Methods). Overall, our mathematical model recapitulates the experimental microfluidics data very well.

## Mathematical model of a senescent population

We next simulated a senescence experiment performed in population (Fig. 1a). Every 24 h, we simulated the dilution of the whole population: we sampled, randomly with equiprobability, a number $N_{dil}$ of cells ($N_{dil} = N_{init}$) to initiate the next daily culture. As soon as the population reached a saturation number $N_{sat} = r_{sat} N_{init}$ (see Methods for details), we stopped making cells divide until the next dilution.

Different laboratories use variations of the same protocol. In the experimental example we used, taken from[24], cultures were diluted to an optical density measured at 600 nm ($OD_{600nm}$) of 0.0125 and were grown in rich media for 24 h. Wild-type strains typically reach 9-9.5 $OD_{600nm}$, which corresponds to ~9.5 cell divisions (Supplementary Fig. 3a)[24,34]. Since wild-type budding yeast divides in 90 min on average in rich media (Fig. 1d), this means that saturation is reached after ~15 h. The culture is then diluted in fresh media back to $OD_{600nm} = 0.0125$.

Based on the above, we simulated cultures of telomerase-negative cells (Fig. 3a). We next simulated curves to compare with experimental population data, specifically the daily monitoring of the cell concentration and the mode of telomere length distribution extracted from[24] (Fig. 3b, c). In this setting, since saturation is reached after ~9.5 divisions, we set $r_{sat} = 720 \approx 2^{9.5}$. We observed in Fig. 3b that simulations of the number of cells and experimental measurement of

**Table 1 | Parameters fitted on the microfluidic data and used for the population simulation experiments**

| parameter | value |
|---|---|
| $a_{nta}$ | 0.02 |
| $b_{nta}$ | 0.44 |
| $a_{sen,A}$ | 0.19 |
| $b_{sen,A}$ | 0.73 |
| $a_{sen,B}$ | 0 |
| $b_{sen,B}$ | 0.12 |
| $\ell_{min,A}$ | 27 |
| $\ell_{min,B}$ | 0 |
| $\ell_{trans}$ | 0 |
| $\ell_0$ | 40 |
| $\ell_1$ | 58 |

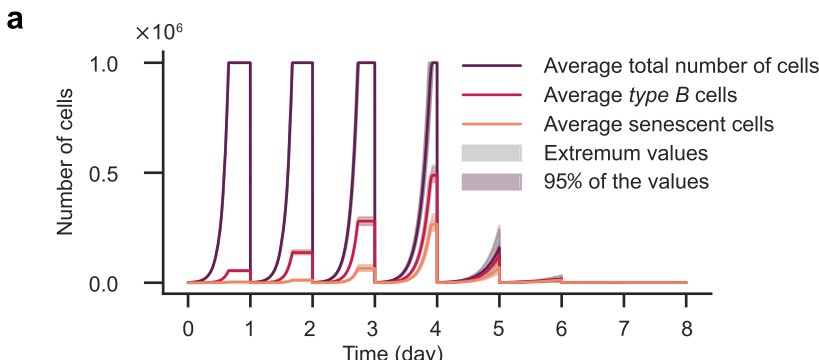

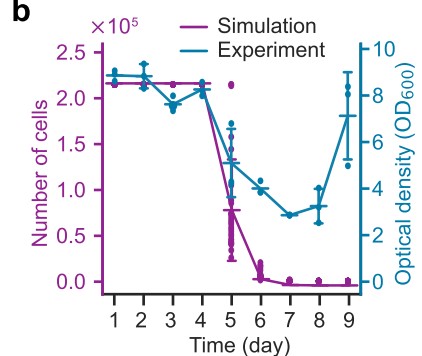

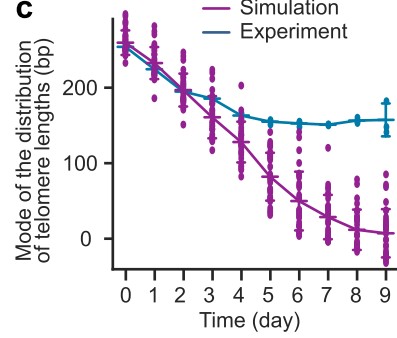

**Fig. 3 | Mathematical model of telomerase-deficient populations. a** Average of 15 simulations of population growth as a function of time, as schematized in Fig. 1a, with best-fit parameters of Table 1 and each day an initial number of cells $N_{init} = 1000$, reaching a saturation number $N_{sat} = r_{sat} N_{init}$, with $r_{sat} = 1000$. Light gray corresponds to extremum values envelope; light magenta area corresponds to 95% of the values. **b** Comparison between experimental data of telomerase-negative cells, taken from[24] and simulations using the best-fit model with $N_{init} = 300$ and $r_{sat} = 720$, corresponding to the experimental conditions (dilution starting at $OD_{600nm} = 0.0125$ and reaching saturation at $OD_{600nm} \approx 9$ after 24 h of growth in

telomerase-positive conditions (see also Supplementary Fig. 3a). To match the experimental plots, simulated values are displayed at the same times as the experimental observations, i.e., once per day. Connecting lines correspond to the mean. Error bars of experimental values correspond to SD of 3 independent experiments. Error bars of simulated data correspond to SD of 30 independent simulations. **c** Comparison of the telomere length mode between the same experiments and simulations as in (**b**). Connecting lines correspond to the mean. Error bars of experimental values correspond to SD of 3 independent experiments. Error bars of simulated data correspond to SD of 30 independent simulations.

$OD_{600nm}$ were in good agreement for the first 4 days, before a drop in proliferation capacity starting at day 5 in both experimental and simulation curves. The same applied to telomere length, for which experimental and simulated telomere shortening fit well, though the discrepancy started at days 3–4. The curves of the experimental growth and telomere lengths were maintained above the simulated profiles from days 4-5 and after. This suggests that our model recapitulates well the events at work in the majority of cells present in the population during the first days spent in the absence of telomerase.

A difference between the experiment and the simulation that may explain the discrepancy is that we started the simulation from 300 cells instead of $\sim 3 \times 10^5$, so that extreme cases have less chance to be part of the initial set. This has two consequences. The first is that we omit from the model post-senescent cells and their descendants, which appear at a low frequency of $-2 \times 10^{-5}$[35]. Accordingly, when we used experimental results obtained in a *tlc1Δ pol32Δ* mutant[7], unable to form telomerase-independent survivors after senescence, the experimental and simulated curves aligned well, suggesting that the exclusion of post-senescence survival in our model is a significant source of the discrepancy observed at later time points (Supplementary Fig. 3c). Post-senescent survivors are expected to contribute to the increase of cell growth and telomere length, but when they initially appear is currently unknown and their emergence could not be detected at the single-cell level in microfluidics experiments level due to their scarcity. Based on the comparison between our simulations and experimental data, they may be already present at day 3 or 4 in a non-negligible proportion. The second consequence is that some extreme cases, like cells having very long initial telomeres, would survive much longer than others and would then also contribute to the population (see below).

Another possible explanation for the difference lies in the method used to experimentally inactivate telomerase (the TetO2 repressible promoter): it might display some leakage, so that a few critically short telomeres could be specifically elongated by very few active telomerase molecules in cells[36]. Similarly, although this may apply to only very few cells, they may become dominant over time when all other cell descendants have entered senescence.

Having these two possible bias sources in mind, we concluded that our simulations best recapitulate the dynamics of a population in the absence of telomerase within the first 3-5 days after loss of telomerase activity. Because it omits possible processes resulting in telomere re-lengthening in a small fraction of cells, our model allows one to dissect the causes and consequences of the mechanisms contributing to the decrease of cell proliferation (i.e., replicative senescence) due to telomere shortening exclusively.

While varying $r_{sat}$ had some minor effect on senescence kinetics (Supplementary Fig. 3d,e), for the rest of the study, we rounded the number of population doublings to reach saturation to 10 ($r_{sat} = 1000$).

## Type B cells gradually replace Type A ones in populations

Our model enables quantification of the heterogeneity of replicative senescence cultures in terms of the composition of cell "age", expressed in generations from the time telomerase was inactivated. As time progressed, the number of generations undergone by cells increased rapidly in early cultures and more slowly in late cultures (Fig. 4a), reflecting a gradual increase in the average cell cycle duration in the population. The variance of these generations also increased substantially with time, reflecting the increasing heterogeneity of cultures. Notably, the proportion of senescent cells became substantial prior to the experimentally measurable decline of cell proliferation in cultures (Fig. 4b). At days 3–4, while one cannot detect a decline in the population proliferation potential (Fig. 3b), 10-25 % of the cells in the cultures are actually senescent according to our model.

The population composition also evolves with time. We observed a progressive replacement of *type A* cells by *type B* cells (Figs. 3a, 4b, c), the latter having in average longer cell division cycles (Fig. 1d), thus

remaining potentially longer in the population and dividing more times (Fig. 1b, c). When we detailed the composition of senescent cells according to cell *types* (*A* or *B*), we observed that on the first day, the tiny portion of cells entering senescence are basically all *type B*, partially misclassified as *type A*, since telomeres of *type A* cells have not yet approached the threshold $l_{min_A} = 27bp$ (Fig. 4d). On day 2, *type A* cells started to enter senescence and from day 3 to day 5 there were significantly more *type A* cells entering senescence than *type B*. The last cells entering senescence from day 5 onwards were mostly *type B*.

We found – and this was expected – that a large fraction of cells (*type A*) entered senescence when the shortest telomere was around 27 bp, which is the $l_{min_A}$ value (Fig. 4d, e). However, for *type B* cells (and *type M*), the length of the shortest telomere was widely distributed, ranging from 0 to 70 bp (comprising ~95% of the values). This will be discussed below.

## Quest for a reliable measurement of population age

Given that the shortest telomeres have been mechanistically associated with the onset of replicative senescence, there is an ongoing debate regarding whether average telomere length or average shortest telomere length serves as the most accurate proxy for biological age[10]. We therefore used our simulations to estimate values that cannot be measured experimentally and evaluated this issue. We plotted the mean and the statistical mode of telomere length distribution in the population over time, and the evolution of the shortest telomere (represented by the average shortest, the shortest shortest, or the longest shortest) in the cell (Fig. 5a). We found that all telomere length values shorten in a non-linear fashion, faster in the early days after telomerase inactivation, compared to late cultures. This was more prominent for the shortest telomere in cells since it is the telomere to which the stronger selection applies. Thus, while the length of the shortest telomere is the major determinant of the onset of senescence at the individual cell level, the telomere length mode acts as a more reliable proxy for the age of a population, as it exhibits a linear-like correlation with time over a longer time scale.

The slowdown of telomere shortening in late culture may reflect the progressive increase in the duration of cell cycles. As cells divide more slowly on average, their telomeres, which shorten with division in our model, shorten more slowly as well. An alternative explanation for the slower rate of telomere shortening in late cultures is the continuous selection of fast-growing cells with longer telomeres. To test this, we clustered the initial cells into 10 equally sized bins according to the length of their shortest telomere, and recorded the number of cells derived from each of these initial cells over time (Fig. 5b–d, Supplementary Fig. 4). We observed that starting at day 3-4, an increasing proportion of the population stemmed from a single population of cells with the longest shortest telomeres. Conversely, the progeny of the cells with shorter shortest telomeres constituted the first senescent cells found in cultures, while the cells having longer shortest telomeres entered senescence later (Fig. 5c). In contrast, as expected, ordering the initial cells by increasing average telomere length revealed a lesser selection effect (Fig. 5d). Among *type B* cells, we found an early selection effect for cells displaying shorter shortest telomere, meaning that although they undergo a non-terminal arrest early in culture, cells may preserve a significant proliferation potential, a tendency that disappears rapidly (Supplementary Fig. 4).

## Initial telomere length distribution influence on senescence

Many mutants and growth conditions affect telomere length homeostasis, and are expected to alter senescence rates[37–41]. To quantify the consequences of altered initial telomere length distribution on the cell population, we ran simulations of populations in which the initial telomere length distribution was modified by a translation of the whole telomere length distribution (see Methods). To estimate the effect on senescence rate in a cell population, we defined the time when only half of the saturation limit is reached (HSL). We found that HSL varied

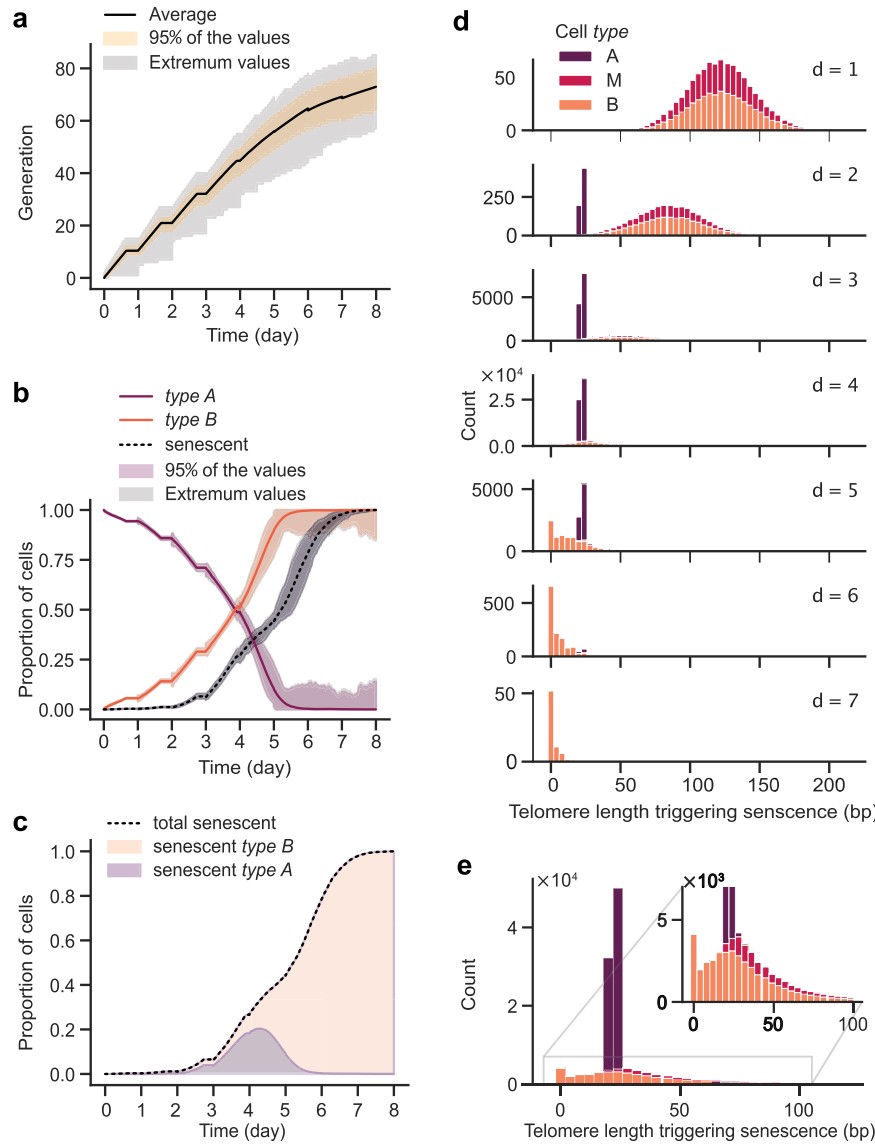

**Fig. 4 | Temporal evolution of cell composition in telomerase-deficient populations. a** Temporal evolution of the generational age distribution of cells. Light gray corresponds to extremum values envelope; Light yellow area corresponds to 95% of the values. **b** Temporal evolution of the proportions of indicated cell categories in the whole population. Light gray corresponds to extremum values envelope; light magenta, gray or light orange areas corresponds to 95% of the values. **c** Temporal evolution of the proportion of type A or type B cells among the senescent population (entering the last set of prolonged cell cycles before cell death). **d** Distribution of the length of the shortest telomere in senescent cells for indicated cell types for each day. **e** Overall distribution of the length of the shortest telomere in senescent cells for indicated cell types.

with the global shift of telomere length distribution in a non-linear way. For example, a translation of -20 bp or +20 bp in telomere length homeostasis led to a change of -14 h or +17 h in HSL, respectively (Fig. 6a and Supplementary Fig. 5a). As expected, average telomere length was shorter over time when initially shorter, and we moreover observe that the shortening rate was maintained for the first 4 days (Fig. 6b and Supplementary Fig. 5b). We then altered the telomere length distribution by stretching out the left-hand side of the distribution by $l_0$, i.e., only the distribution of the shortest telomere lengths. Because we altered only the distribution of the shortest telomeres, the average telomere length of the population over time remained mostly unchanged (Fig. 6d and Supplementary Fig. 5d). However, this had unexpectedly only minor effects on senescence rate (i.e., HSL, Fig. 6c and Supplementary 5c). This counterintuitive result can be explained by the pronounced selection pressure acting on cells, where those possessing longer shortest telomeres remain largely unaltered following this transformation. Hence, as long as a population

contains a few cells with long shortest telomeres, it will consistently demonstrate proliferation capacity.

**Spontaneous cell mortality accelerates senescence**

To mimic the context of numerous pathological conditions associated with increased cell turnover and renewal, or cells and tissues exposed to a cytotoxic compound, we evaluated the impact on senescence of a perturbation that is not directly linked to telomere processing by increasing the telomere-independent and constant mortality. We found a dramatic effect on senescence curves, with an anticipated HSL of -1 day at 20x increase in mortality, which corresponds to an increase from 0.43% estimated in wild-type cells, to 8.6% (Fig. 7a and Supplementary Fig. 6a). Also, the apparent shortening rate in the population was found to increase significantly with increasing constant mortality (Fig. 7b and Supplementary 6b). Thus, the kinetics of replicative senescence and telomere shortening at population level are strongly influenced by co-morbidities. This effect is also visible when using the

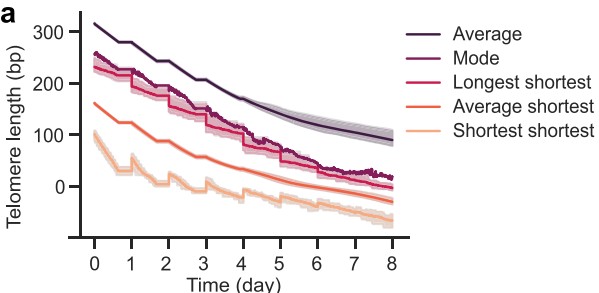

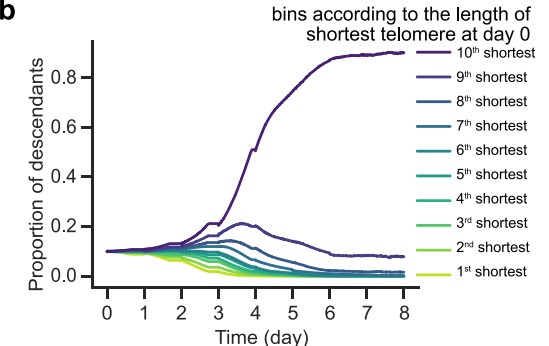

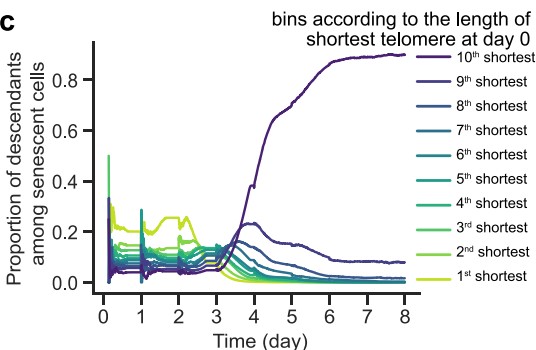

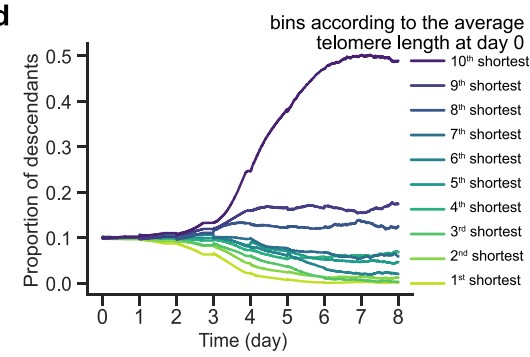

**Fig. 5 | Hidden parameters of telomerase-deficient populations that are experimentally inaccessible. a** Temporal evolution of indicated telomere length distribution features. Gray error bands represent the extremal values, while the other error bands encompass 95% of the values (+/− 97.5%/2.5% percentiles). **b** Proportion of descendant cells in the population based on the initial shortest telomere length of their ancestors prior to telomerase inactivation. **c** Same as (**b**) for senescent cells only. **d** Proportion of descendants based on the average telomere length of their ancestors prior to telomerase inactivation.

model applied to single lineages to simulate senescence in microfluidics experiments (Fig. 7c, d).

Many mutants that affect senescence are assumed to function in telomere maintenance, but they may affect senescence indirectly by, for example, promoting cell death. To test whether our model can distinguish between these mechanisms, we examined the deletion of *RAD51*,

which encodes a recombinase implicated in homologous recombination. Rad51 also binds to short telomeres and is involved in the emergence of post-senescence survivors, and perhaps other mechanisms operating at short telomeres[35,42–44]. *RAD51* deletion accelerates senescence in cell populations[42,43,45,46]. However, it also increases the constant mortality to ~5.4%, probably due to other functions in double-strand break repair or replication stress. To measure the contribution of the constant mortality caused by *RAD51* deletion in the observed accelerated senescence of *rad51Δ* in telomerase-inactivated conditions, we performed 1000 simulations of 11 individual cell lineages as if they were grown in the microfluidics device in the absence of telomerase and with a constant mortality rate of 5.4% and compared it to experimental data extracted from[18]. We found that a large portion of the experimental curve fit within the confidence interval of simulations (Supplementary Fig. 6c). A similar result was found when populations of telomerase-negative *rad51Δ* cells were simulated and compared to experimental results (Supplementary fig. 6d). We conclude that accelerated loss of proliferation in telomerase-negative *rad51Δ* cells is largely caused by telomerase-independent Rad51-dependent constant mortality both at the lineage and population levels. However, because part of the experimental curves lies outside the confidence intervals, Rad51 may specifically affect cells also when telomerase is absent, for instance by promoting homologous recombination at short telomeres and contributing to survivor emergence. Hence, assessing the synthetic lethality of *RAD51* deletion using replicative senescence experiments as a readout may not be the most suitable approach for evaluating the genetic interaction between telomerase and the *RAD51* gene.

## Discussion

In this study, we leveraged microfluidics-based data on individual lineages of dividing yeast cells with inactivated telomerase to build a comprehensive model of replicative senescence, and then applied it to simulate yeast population assays commonly used to study replicative senescence. We unified, tested, and validated information from previous separate studies in which we found that cell divisions are constrained by telomere shortening, and more specifically by the shortening of the shortest telomere in cells, and not simply by time or other telomere-related parameters, to fully account for replicative senescence heterogeneity in *S. cerevisiae*[20]. It is known that selection biases occur between lineage and population observations[47–49]. Simulation enables drawing rigorous conclusions on population dynamics from lineage data observations. Our study contributes to this methodology, and allows us to decipher which phenomena and mechanisms prevail in populations, sometimes quite distinct and even counter-intuitive compared to what might be assumed at first sight from lineage data.

One of the most striking findings of this model lies in the contrasted route to senescence of *type A*, for which senescence occurs at an almost deterministic threshold of around 27 bp for the shortest telomere, versus *type B* cells, for which the probability to trigger senescence is telomere-length independent after experiencing a priming telomere-length-dependent non-terminal arrest (Fig. 8). We thus suggest the existence of tight control mechanisms responsible for triggering senescence when a telomere becomes critically short in *type A* cells. Conversely, the mechanism signaling the terminal arrest in *type B* cells is likely distinct from the one operating in *type A* cells. *Type B* cells could die from indirect consequences of some processes initiated during the non-terminal arrests, either the prolonged DNA damage checkpoint activation and its genetic and epigenetic reprogramming, or the mutational burden associated with it.

In a budding yeast experimental setting where one telomere is set shorter than the others and specifically sequenced, telomeres with lengths between 10 and 70 bp can be extracted from a senescing cell population, and even some chromosome ends lacking telomeric repeats can be detected[19]. Recent experimental works measuring telomere length in telomerase-negative cells using long-read sequencing

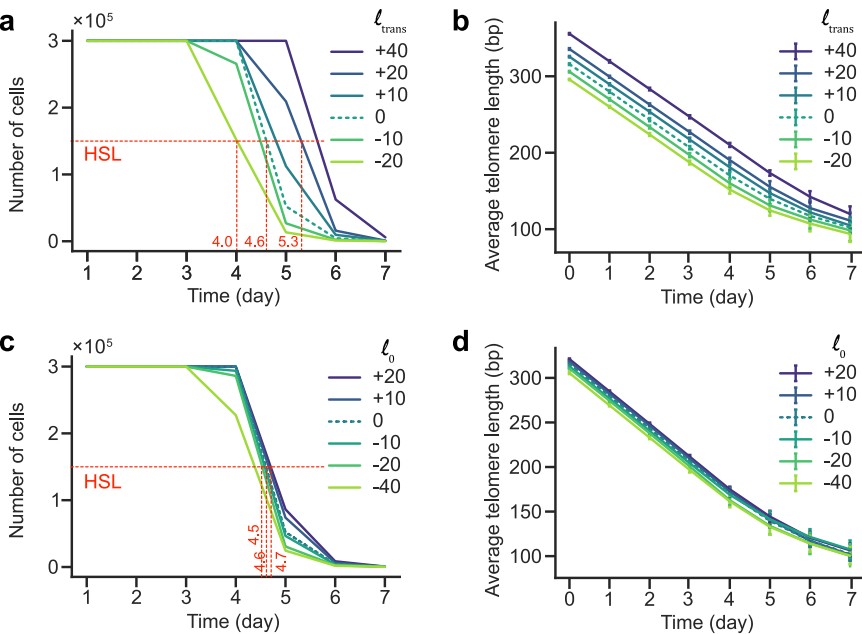

**Fig. 6 | Effects on senescence rates from altering telomere length prior to telomerase inactivation.** Plot of the simulation of population growth (**a**, **c**) and linked average telomere length (**b**, **d**) as measured each 24 h, with best-fit parameters of Table 1 and each day an initial number of cells $N_{init} = 1000$, reaching a saturation number $N_{sat} = r_{sat}N_{init}$, with $r_{sat} = 1000$. Half of the saturation limit (HSL) and relevant x-axis coordinates are indicated in red. **a**, **b** Effect of initial global telomere length distribution translation towards longer or shorter average telomeres on replicative senescence at the scale of populations. **c**, **d** Effect of altering positively or negatively the left side of telomere length distribution (the shorter telomeres). **b**, **d** Data are presented as mean values +/− SD.

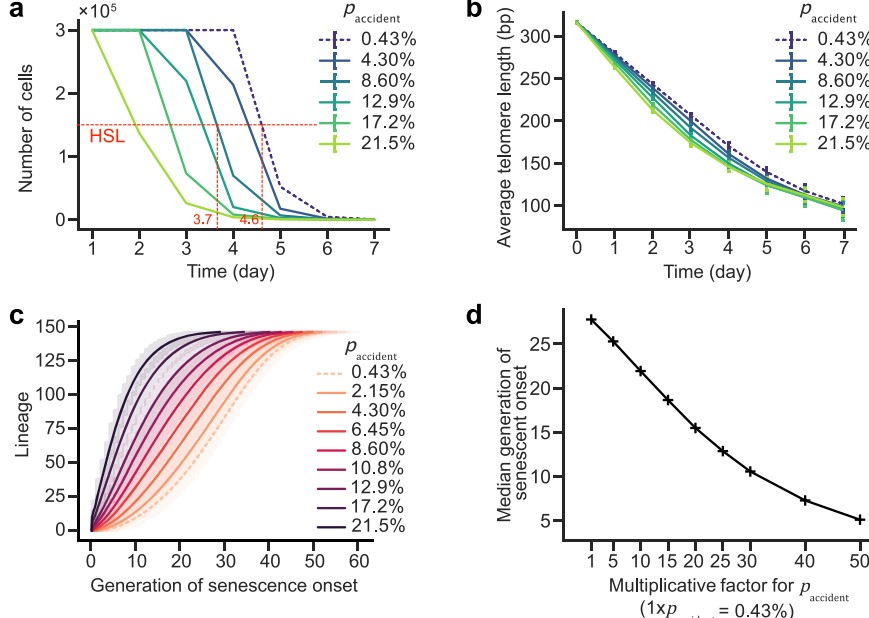

**Fig. 7 | Effects on senescence rates from altering telomere-independent spontaneous mortality rates ($p_{accident}$).** Plot of the simulation of population growth (**a**) and linked average telomere length (**b**) as measured each 24 h, with best-fit parameters of Table 1 and each day an initial number of cells $N_{init} = 1000$, reaching a saturation number $N_{sat} = r_{sat}N_{init}$, with $r_{sat} = 1000$. Half of the saturation limit (HSL) and relevant x-axis coordinates are indicated in red. **b** Data are presented as mean0 values +/− SD. **c** Proliferation of individual cell lineages of strains displaying indicated constant mortality rates ($p_{accident}$) as simulated by single-lineage model, i.e. grown in the microfluidics device. Generation of senescence onset was ordered by lineage lifespan. Gray error bands represent the extremal values, while the other error bands encompass 95% of the values (+/− 97.5%/2.5% percentiles). **d** Median generation onset derived from (c) plotted as a function of increasing ($p_{accident}$).

proposed an estimate of the threshold length for the critically short telomere around 70–75 bp[50]. When we take into account both *type A* and *B* cells in our model, we find an average length lower than 70-75 bp (Fig. 4e). However, we note that this average, and in particular the threshold length of 27 bp for *type A* cells which contributes to it, should

not be taken as an exact value, but rather as an estimate which can depend on simplifying assumptions of the model, on the data and the parameter estimation method used to fit, and, above all, on the distribution of telomere lengths used to initiate the simulations. The latter has indeed a strong influence on the threshold $l_{min,A}$ and suffers

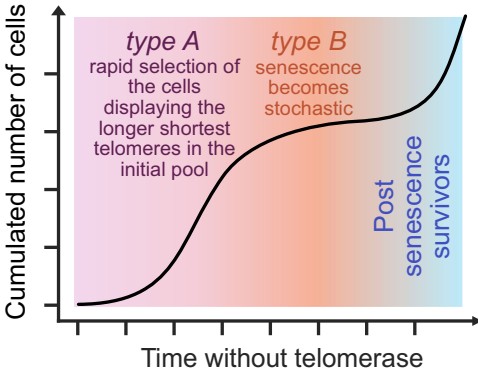

**Fig. 8 | Major features of the temporal evolution of telomerase-deficient cell populations according to our mathematical model of replicative senescence.** The cell composition of a telomerase-deficient population evolves over time. Initially, the population consists of cells that enter senescence when their shortest telomere reaches a critical length. As these cells become exhausted and the population growth slows, cells experiencing non-terminal arrest begin to accumulate, as their frequency increases with telomere shortening. These cells enter senescence in a manner independent of the length of their shortest telomere. We speculate that these cells could potentially become post-senescence survivors, leading to renewed population growth.

from approximation. It is important to note that for *type B* cells, we observed an accumulation of cells entering senescence at $l_{\min,B} = 0$, indicating that in vivo, *type B* cells might indeed be the carriers of telomere-free chromosome ends. We also refer to the works of[50–52], which suggest that individual chromosome-end-specific telomeres display differences in homeostasis length (for example, the telomere of chromosome arm 3L is consistently longer than the others in *S. cerevisiae* strains), possibly giving rise to different critical length thresholds for senescence. Furthermore, subtelomeric elements and heterochromatin status *in cis* might affect the critical length. Thus, the threshold for critically short telomeres is likely not a single value but a set of values, each specific to a chromosome end. Our simulations also indicate that the average shortest telomere length, as well as the average threshold length for senescence decreases over time because of competition and, as a consequence, the change of population structure, i.e. relative proportion of *type A* and *B* cells (Fig. 4d).

When simulating population assays, we did not take into account post-senescence survivor emergence for two reasons: first, no microfluidics-based data was available to build and calibrate a model. Second, we wanted to focus on understanding to which extent replicative senescence on its own could explain experimental observations. Our study highlighted the presence of heterogeneity within the population, where cells of different generational ages and different histories co-exist, in particular *type B* cells, which have potentially experienced molecular and cellular events leading to genomic instability[27]. Importantly, our observation that senescent cells are mainly *type A* between day 1 and 4, then after day 4, mainly *type B* (Fig. 4b, d) – an alternate dominance well-explained by their contrasted route to senescence – reinforce the idea that *type B* cells might be poised to generate post-senescence survivors. In this scenario, it is possible that the non-terminal arrests could correspond to attempts at DNA repair operating at the signaling telomere that could alter the length and structure of the shortest telomere itself, as suggested[53]. Alterations in telomere length distributions in *type B* cells during non-terminal arrests—such as the potential re-lengthening of the shortest telomere—could account for the emergence of survivors and the shift in senescence behavior, which becomes telomere-length independent. However, to incorporate post-senescent survivors into our model, more precise mechanistic insights are required. For instance, it is currently unknown whether the shortest telomere is the first to be

elongated and, if so, to what length. Additionally, it is unclear whether these events occur with or without cell cycle arrest.

Beyond the competition affecting population structure and dynamics, shifting perspective from individual lineages to populations reveals that, while the length of the initial shortest telomeres strongly constrains senescence onset in lineages, the entire initial distribution of telomere length is important for population growth dynamics (Fig. 6a, c). This is in accordance with the strong selection bias towards cells displaying the longest shortest telomeres in the initial population (Fig. 5b–d, Supplementary Fig. 4). We can also speculate that as time passes, senescence in a population is more often triggered by the telomeres that were initially ranked as second, third, fourth, etc. shortest telomere in the ancestor cell and later became the shortest[20]. These dynamics can be explained by the asymmetry of telomere replication, yielding two newly-replicated telomeres of different lengths, which would in some lineages lead to changes in the initial ranking of telomeres, as we previously simulated[20]. A direct consequence of this dynamics is that the size and telomere length heterogeneity of the initial population prior to telomerase inactivation is critical to predict proliferation capacity. Conversely, if a specific critically short telomere is inherited in the same way in all cells, it is expected to have a significant impact on the whole progeny, in accordance with experimental results[15,19]. In the context of humans, for instance, where telomerase activity is often repressed in somatic tissues, the genealogy of telomerase-positive stem cell compartments would thus be crucial. Schematically, a small and closely related stem cell compartment, possessing similar telomere length distributions, would exhibit a senescence rate highly dependent on the length of the inherited shortest telomere. Conversely, a larger population with widely distributed telomere lengths would harbor more cells with relatively longer shortest telomeres, thereby sustaining the tissue's proliferation and renewal capacity for a longer duration.

Replicative senescence in the absence of telomerase has been extensively investigated in different genetic backgrounds and various conditions. Conditions where accelerated or delayed replicative senescence is observed, by comparing the kinetics of senescence at the population level, have often been interpreted as interfering with telomere biology[30,43,54]. However, the fact that replicative senescence displays an intrinsic heterogeneity and a non-constant mortality makes it not trivial to draw such conclusions, as exemplified by the *RAD51* deletion effect on senescence, which we suggest to stem in great part from its intrinsic mortality and only to a lesser degree from a specific requirement of Rad51 in the absence of telomerase, for instance by promoting homologous recombination at short telomeres and contributing to survivor emergence (Supplementary Fig. 6c, d). In addition, relying solely on telomere length at specific time points or telomere shortening as predictors of senescence is inadequate, particularly when the initial telomere length is modified or mortality is influenced by factors other than telomerase inactivation.

While the existence of *type B* cells remains to be established in other species than *S. cerevisiae*, our mathematical model built based on a unicellular eukaryote paves the way for in-depth exploration of the intricate relationship between telomere length, shortening dynamics[9], and cell growth in different tissues in metazoans (as in ref. 55). Aspects such as genomic instability, shown to raise in the course of replicative senescence[56–59], and expected to impact fitness could also be addressed in future work, combining our mathematical model with others[60,61]. Our mathematical model might thus serve as a valuable approach for investigating the broader implications of telomere length dynamics combined with co-morbidities and genome instability.

## Methods
### Telomere shortening model
We use the model of[20], which consists in imposing that for each chromosome 1/ Only one of the two telomeres is shortened, the other conserves the parental length; 2/ It is shortened by the overhang, say h,

assumed constant; 3/ The telomere shortened for one daughter is unchanged for the other daughter. Mathematically, at the $n$-th generation, we denote by $l_1^n$ and $l_2^n$ the random variables of the lengths of the two telomeric ends of a given chromosome at generation $n$. One and only one telomere is shortened by the overhang $h$ with equiprobability, such that at the next generation for one of the two daughters we have $l_1^{n+1} = l_1^n - hb$ and $l_2^{n+1} = l_2^n - h(1-b)$, and for the other daughter it is the reverse: $\underline{l}_1^{n+1} = l_1^n - h(1-b)$ and $\underline{l}_2^{n+1} = l_2^n - hb$, where $b \sim Ber(1/2)$ is a Bernoulli random variable coupling the two telomere lengths ($l_2^{n+1}$ is shortened by $h$ nucleotides if $b$ is $0$, while $l_1^{n+1}$ is left unchanged, and conversely if $b$ equals $1$). Inside cells containing $2k$ telomeres ($k = 16$), we assume that telomeres of different chromosomes are independent[62], so that denoting by $L^n = (L_1^n, L_2^n)^T$ the matrix of size $2xk$ of the lengths of the $2k$ telomeres, we have similarly for the two daughters $L^n$ and $\underline{L}^n$

$$\begin{cases} L_1^{n+1} = L_1^n - hB, & \overline{L}_1^{n+1} = L_1^n - h(1-B), \\ L_2^{n+1} = L_1^n - hB, & \overline{L}_2^{n+1} = L_2^{n+1^n} - h(1-B), \end{cases}$$

with $B = (B_1, \ldots, B_k) \sim Ber(k, 1/2)$ is a random vector of $k$ independent Bernoulli variables.

To model the population experiment (Fig. 1a), we keep the two daughters at each division, whereas for the microfluidic experiment (Fig. 1b), we pick up one of the two matrices $(L_1^{n+1}, L_2^{n+1})$ and $(\underline{L}_1^{n+1}, \underline{L}_2^{n+1})$ randomly uniformly.

## Initial distribution of telomere lengths

We assume that initially all telomere lengths are independent identically distributed (*i.i.d.*) according to a law $f_{init} : L_{i,j}^0 \sim f_{init}$, $i \in \{1,2\}$, $j \in \{1, \ldots, 16\}$. Given that generation $0$ corresponds in our dataset to the inactivation of the telomerase, we depart from the distribution of telomere lengths in a telomerase-positive population (of the same yeast strain as the dataset) at equilibrium. We rely on the distribution of telomere lengths $f_0$ of[20] (Supplementary Fig. 1b) derived by adapting the numerical approach of ref. [15] to this yeast strain. Given that the left-tail of the distribution has great influence on the lineages dynamics but is poorly characterized experimentally, we test small modifications of $f_0$, namely translation (by a given length $l_{trans}$) and dilatations preserving the mode of the distribution, i.e. dilatations defined by dilating $[l_{inf}, l_{mode}]$ (resp. $[l_{mode}, l_{sup}]$) to $[l_{inf} + l_0, l_{mode}]$ (resp. to $[l_{mode}, l_{mode} + l_1]$) (see Supplementary Material for further detail). We optimize the values for $l_0$ and $l_1$ together with the other parameters of the model, see Table 1 for the optimal values and Supplementary Fig. 1b for the initial distribution $f_{init}$.

## Laws of arrest

Once its telomere lengths determined by the telomere shortening model, the cell type is chosen according to the transition probability laws described by the tree diagram of Fig. 1e. The probability rate $p_{accident}$ represents the chance to die accidentally and is constant, taken from[27]. The rates $p_{sen,A}$ and $p_{sen,B}$ are the probability to enter senescence respectively for *type A* and *type B* cells. They depend on the minimal telomere length $l = \min(L^n)$ through the law

$p_{sen,i}(l) = \min(1, b_{sen,i} \exp(-a_{sen,i}l))$ if $l > l_{min}$, $\quad p_{sen,i}(l) = 1$ if $l \leq l_{min}$, for $i = A$ or $B$.

These laws are thus defined by five parameters ($a_{sen,A}$, $b_{sen,A}$, $a_{sen,B}$, $b_{sen,B}$, $l_{min}$). Figure 2a displays their best-fit values, which shows a remarkable fact: we can simplify the law for *type A* cells into an all-or-nothing law $p_{sen,A} = 0$ for $l > l_{min}$ and $p_{sen,A} = 1$ for $l \leq l_{min}$, whereas $p_{sen,B}(l) = b_{sen,B} \exp(-a_{sen,B}l)$ is a very flat almost uniform law, so that finally only three parameters matter, namely $l_{min}$, $a_{sen,B}$ and $b_{sen,B}$.

The rates for non-terminal arrests are similar: we define them through the same two-parameter law as for senescence: $p_{nta}(l) = \min(1, b_{nta} \exp(-a_{nta}l))$. Finally, when a cell experiences a

sequence of arrests[28], has successfully described the number of consecutive abnormally long cycles, either terminal or not, by a geometrical law. This corresponds to a constant probability to exit this sequence either by repairing or adapting ($p_{repair}$) after a non-terminal arrest, or at the opposite by dying ($p_{death}$) after a senescent cycle.

## Laws of cell cycle duration times

Whereas cell cycle durations are not fundamental to simulate the microfluidic experiments, they are crucial to model the population experiment, where the evolution axis is no longer generation but time. Based on experimental measurements, and on a threshold $D$ above which a cell cycle duration is considered abnormally long, we separated the microfluidic dataset (Fig. 1c) into four subdatasets corresponding to four distinct regimes in the model: normal cell cycle durations for *type A* and *type B* cells, non-terminal arrest cycle durations and senescent cycle durations (Supplementary Methods Fig. 1a; or Fig. 1d for kernel density smoothing representation). According to the cell type and cell cycle type, the cycle duration is then picked up at random, according to a uniform law, in the associated subdataset.

## Parameters values

**Parameters from the literature.** The shortening length is taken as the overhang length $h = 7$ bp, see ref. [24]. We define the threshold for abnormally long cell cycle durations $D = 180$ min, following the thorough sensitivity analysis carried out in ref. [28]. From the same reference, we fixed $p_{death} = 0.58$ and $p_{repair} = 0.65$. From[27] we get $p_{accident} = 4.3 \times 10^{-3}$.

**Parameters from direct experimental measurements.** In our experiments we have measured $r_{sat} = 720$. We also pick up at random the cell cycle durations from the experimental distributions, classified into the four categories displayed in Fig. 1d.

**Parameters inferred from the microfluidics experiments.** We infer 10 parameters from the microfluidics experimental data, namely: ($a_{nta}, b_{nta}, a_{sen,A}, b_{sen,A}, a_{sen,B}, b_{sen,B}, l_{minA}, l_{minB}, l_0, l_1$). We use the CMA-ES algorithm to minimize the distance between the experimental graphs of Fig. 2b and several repeats of "simulated experiments": examples of simulated experiments are displayed in Fig. 2c and Supplementary Fig. 2a, b. The distance is defined by a certain cost function, see Extended Methods for its formula.

**Determination of $N_{init}$.** In terms of cell quantities, the population experiment corresponds to a growth from ~375 000 cells/ml up to ~3 × $10^8$ cells/ml each day, unless otherwise stated. Simulating the exponential growth of such a high number of cells is too expensive, in terms of computation time and allocated memory, to keep track of all the parameters of the cells (Supplementary Fig. 3b). We thus tested whether simulating smaller populations generated a bias. To this aim, we identified a minimal size $N_{init} = 300$ at which populations were big enough to account for the experimental data in terms of cell growth. We then defined a saturation ratio $r_{sat}$ in order to stop the simulation each day when the number of cells reaches $N_{sat} = r_{sat} N_{init}$.

To determine $N_{init}$, we compare the behavior of populations originating from different initial number of cells $N_{init}$. In order to accurately estimate these behaviors (i.e. to have an empirical behavior close to the statistical one) we simulated 25 times the evolution of a population with a certain fixed initial size $N_{init}$. The full sensitivity analysis is detailed in Supplementary Material, see also Supplementary Methods Fig. 2. We first noticed that the less cell initially present, the more variability between simulations, which supports the idea that the variability in the distribution of initial telomere lengths is an important source of heterogeneity in senescence[15,19]. Even though extreme behaviors are sensitive to $N_{init}$ unlike average behaviors, the decrease in variability stabilizes around $N_{init} = 200$, and then corresponds to the variability intrinsic to the stochastic evolution.

**Yeast strain and methods.** Two independent subclones of two independent *rad51::LEU2* transformants of *Mat* **α** *ura3-1 trp1-1 leu2-3,112 his3-11,15 cdc10::CDC10-mCherry-kanMX tlc1::HIS3MX6-PrTetO2-TLC1* of a *ADE2 RAD5*-corrected W303 strain (yT641[18]) were inoculated at $OD_{600nm} = 0.0125$ in YPD containing 30 μg.ml[-1] of doxycycline, grown at 30 °C for 24 h, and final OD was measured. This was repeated for 9 consecutive days and the whole experiment was repeated once to obtain 8 independent senescence curves.

## Reporting summary

Further information on research design is available in the Nature Portfolio Reporting Summary linked to this article.

## Data availability

The data generated in this study is available at https://github.com/anais-rat/telomeres[63]. Source data are provided with this paper.

## Code availability

The code use for simulation, parameter estimation and figure plotting in this study is available at https://github.com/anais-rat/telomeres[63].

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

## Acknowledgements

We thank Jim Haber for sharing raw data of published results. We also thank Claus Azzalin, Miguel Godinho Ferreira, Andrew Paek, Ted Weinert and Helen Pickersgill from Life Science Editors for critical reading of the manuscript. This project was supported by the ERC Starting Grant SKIPPERAD 306321 (AR, MD), ANR-16-CE12-0026 (MD, MTT, ZX), the "Institut National du Cancer" INCa_15192 (MD, MTT, ZX), the "Fondation de la Recherche Medicale" (MTT), the "Investissements d'Avenir" Program LabEx Dynamo ANR-11-LABX-0011-01 (MTT), and the Mairie de Paris "Programe Emergences" (ZX).

## Author contributions

Conceptualization, Funding acquisition, Supervision: M.D., M.T.T., ZX; Methodology: A.R., M.D., Z.X.; Software: AR; Investigation & Writing: A.R., V.M.F., M.D., M.T.T., Z.X.

## Competing interests

The authors declare no competing interests.
