## [Transparent Peer Review file · Nature Communications]

Mathematical model linking telomeres to senescence in *Saccharomyces cerevisiae* reveals cell lineage versus population dynamics

Corresponding Author: Dr Maria Teresa Teixeira

Version 1:

Reviewer comments:

Reviewer #1

(Remarks to the Author)

The paper is devoted to a comprehensive study of telomere dynamics in yeast cells, which is based on experimental data from cell cultures with daily passages and cell lineage statistics obtained from microfluidics. The idea is that combining these two types of data, it is possible to uncover detailed information about temporal trends in aging yeast cells, with possible implications for senescence of human cells, and with specific reference to carcinogenesis.

The simulation model is a complex multitype branching process with the cell types identified as being (i) the lengths of telomeres in individual yeast cells, and (ii) the cell types A and B, with the A cells inevitably dying after becoming senescent, and B cells capable of returning from senescence (with appropriate transition rules as in Figure 1, and rules related to cell cycle lengths). An additional M-type of cells seem to be B cells, indistinguishable experimentally from the A cells (see further on).

One of the most interesting phenomena observed in the experiments is temporarily reversible senescence in B-type cells, which occurs despite the continued shortening of telomeres. Many, although not all (see Figure 3 B and C), characteristics of experimental data are replicated, and some characteristics which are unobservable experimentally are strikingly illustrated by the simulations (Figure 5).

All in all, this may be the most comprehensive and realistic mathematical model of telomerase-negative cell senescence in the literature. In my opinion it is of sufficient quality to be published in the Journal, provided certain questions are addressed in a revision.

Detailed comments:

1. I think it will be useful if the authors discuss why the A cells cannot simply be a variety of B cells with 0 returns from senescence. There may be a biological reason that I do not see, though.
2. Line 83. It is quite interesting how the "telomere decay" genomic instability fits into a wider picture of genome duplication followed by chromosome mis-segregation (see papers by Watkins et al. in Nature or Dinh et al. in bioRxiv <https://doi.org/10.1101/2024.04.03.587939>). A connection certainly exists (see eg. Lo et al. Neoplasia 4.6 (2002): 531-538 and papers citing this one). Can any details be elucidated by this model?
3. Line 263. "Type B gradually replace (replaces?) Type A in senescent populations". I would risk saying that this observation very simply follows from a mathematical representation of the fates of the B cells as a series of failures the number of which is described by a roughly geometric distribution (thus with mean count of failures > 1), while the A cells live only through 0 failures.
4. Line 277. The M cells. Are they B misclassified as A, or simply A and B are the same cells with different random count of returns from senescence. And, could you explain what "misclassification" means, using the example of a specific cell observation?

5. Lines 398-399. Might you elaborate on these differences?

6. Line 422-424. "We can also speculate ... is more often triggered by the telomeres that were initially ranked as second, third, fourth, etc. shortest, etc." You cite some figures in support, but maybe a more detailed discussion of this interesting hypothesis is in place? What is the model-based evidence for why and how would this happen?

Minor issue

Line 32. "how factors spuriously influencing cell viability". What does "spuriously" mean in this context?

(Remarks on code availability)

More specifically, I reviewed the code annotation, which seems to be sufficiently transparent. The link provided works after it is pasted into my Mozilla browser.

Reviewer #2

(Remarks to the Author)

The reviewed manuscript by Rat et al, describes a novel and original approach for mathematical modeling of growth and senescence in a population of yeast cells undergoing telomere shortening upon the depletion of telomerase. Cells in the culture are experiencing replicative senescence, non-terminal growth arrest and accidental telomere length-independent senescence. The model takes into account these processes and describes the growth and telomere dynamics of individual cells and lineages, as well as the growth on the population level, based on experimental data from single-cell lineages in microfluidics experiments. Mathematical simulations of individual cell lineages show almost identical results to the microfluidics experiments, validating the mathematical model. On the population level the model fits well the experimental data for the first 3 days upon depletion of telomerase, while later on the experimental data (but not the simulated) shows the recovery of the population, possibly due to selection of a subset of cells that bypass senescence. The authors used the mathematical model to test the contribution of the shortest telomeres, the mode and heterogeneity of telomere length, and morbidity unrelated to telomere length, to the lifespan of cell lineages and population dynamics. These experiments highlight the (under-appreciated) contribution of telomere length heterogeneity, founder effect of short telomeres, and telomere length-independent morbidity to senescence. I find these results extremely interesting and important to our understanding of population dynamics and senescence, not only in unicellular organisms such as yeast, but also in human somatic tissues, which are mostly telomerase negative. Such a model applied to human cells can help understanding the effects of compromised telomerase activity in telomere-related diseases, cancer, and clinical conditions that demand increased cell proliferation (e.g., bone marrow transplantation). It can also elucidate the contribution to senescence of increased telomere length heterogeneity and the combination of short telomeres and telomere length-independent morbidity, as observed in various clinical conditions. While beyond the scope of this manuscript, the mathematical approach reported here will hopefully provide the basis for such more complicated simulations of human cells in the future. The manuscript is clearly written and the conclusions and biological implications of the mathematical model are sound and well articulated.

Comment:

The main issue that puzzles me is whether recombinational repair that elongates the telomeres can explain (1) the emergence of survivals and population recovery in the experimental data, and whether this can also be modeled; and (2) the behavior of type B cells, which undergo terminal senescence in a presumed telomere-length independent manner. If recombinational repair is initiated in a temporary manner upon the non-terminal arrest of type B cells, perhaps telomeres are already elongated at this stage. Then the repair pathway is silenced and telomeres resume shortening till reaching senescence at the same telomere length as type A cells. These are of course speculations, but I'd be happy if the authors elaborate on this issue a bit more. Or otherwise, what is the genetic-molecular difference between type A and type B cells? See also below.

Specific comments:

1. Lines 192, 197: There is no figure 1G.

2. Lines 241-243: This sentence can be simplified, for example: "A difference between the experiment and the simulation that may explain this discrepancy is..." On line 251, I suggest "Another possible explanation for the difference...", and on line 256: "these two possible bias sources..."

3. Lines 242 and 244: $\sim 3 \cdot 10^5$ and $\sim 2 \cdot 10^{-5}$, do the authors mean 3×10^5 and $\sim 2 \times 10^{-5}$? May be clearer to the broad readership.

4. Lines 243-244: Have the authors tried to add to the simulation also the emergence of post-senescence survival at the indicated rate measured experimentally ($\sim 2 \cdot 10^{-5}$)? Perhaps it can decrease the discrepancy? Or, compare experimental versus simulated population growth (as in Fig. 3B,C) for the strain lacking RAD51 (shown in Fig. 7E)?

5. Lines 319-323 and Figs. 6A and S5A: Note that at days 4-6, when the effects of shifting the telomere length on HSL are apparent, there is a large discrepancy between the simulated and experimental results. Please address the relevance of the simulation here.

6. Lines 358-362: The authors performed simulation of a strain lacking RAD51 with increased constant mortality, and observed a discrepancy between experimental and simulation (Fig. 7E). Could this discrepancy be explained by the role of RAD51 in recombinational repair and the generation of type 1 post-senescence survivors? Please clarify better the last two sentences of this paragraph.

7. Type B cells are simulated to enter senescence at a telomere length-independent manner, and also accumulate senescing cells without a telomere at all ($l=0$). I wonder if this result together with the simulation of the rad51 mutant are not suggesting that type B cells can elongate their telomeres at the non-terminal senescence state (lines 412-415) by recombinational repair and therefore enter terminal senescence at telomere lengths that are actually longer and less heterogenous than the short and heterogenous telomeres simulated without taking into account recombinational repair.

8. Lines 387-389: If experimental measurement by long-read sequencing identified 70-75bp as the length of critically short telomeres, does it make sense to use this value to correct the mathematical model?

9. Figure 1C: Can the authors enlarge the examples shown for single lineages?

10. Figure 3A: Difficult to see the extremum and 95% values. Perhaps they are not important here and can be removed from this figure?

11. Figure 5A: by Shortest, do you mean 'Shortest shortest'?

12. Figure 5B-D: Perhaps it would help the reader to reverse the order of the bins listed on the right (i.e., 10th on top and 1st on the bottom).

13. Figure S3B: What are the units Mo?

(Remarks on code availability)

Reviewer #3

(Remarks to the Author)

In their manuscript, Rat et al. present a mathematical model for population dynamics in telomerase-negative yeast based on replicative senescence driven by telomere shortening. The model integrates the research group's previously generated insights, is calibrated on existing experimental data, and yields several findings regarding e.g. the impact of the shortest telomere. I am convinced that this type of mathematical modelling is essential to provide mechanistic insight into how (critical) telomere shortening at a cellular level impacts the larger scale, yet I have several major concerns regarding this study and its presentation.

A first major concern is that I am currently not convinced about the relevance of the model (and hence the authors' results). Contrasting the title making a generic claim, this study focuses solely on the population dynamics of telomerase-negative yeast. Even if the model would successfully capture these dynamics, it is unclear to which extent findings can be biologically extrapolated to any other setting, even telomerase-positive yeast for that matter. Though I can imagine that these population dynamics partially reflect the population dynamics of cells in tissues of metazoans affected by critical telomere shortening (such as humans), there are simply too many differences to consider the first an accurate model for the latter. For example, the role of (telomerase positive) stem cells/progenitor cells, the role of sampling when culturing yeast, the impact of contact inhibition in metazoans, the question whether "type B" is also relevant beyond yeast etc. Note that I fully agree that yeast is a valid model system for telomere length biology and dynamics at the cellular level, but that I oppose to claims of generic relevance (as suggested by the title) at higher levels. I therefore also suggest to clarify the exact aims of this research.

Additionally, the work is currently presented in a very opaque way. I am well aware of the fact that presenting novel, complex models is not straightforward, as it requires both a mathematical description and a biological rationale for the description. Typically, in the results section the model and its rationale are introduced, and the methods section explains the implementation, parameter estimation, computational workarounds etc. In this manuscript, however, the results section continues to elaborate on how the model was fit to data, and why there are certain discrepancies with experimental data, with only minimal attention to what the assumptions or results imply from a biological point of view. E.g. the results section states that they started the simulations with telomere lengths randomly drawn from the distribution described in 12,17 (lines 157/158), a few lines further (163/164): "the length of the shortest telomere in cells is computed after running the telomere shortening model described in 19" etc. Solely providing references without mentioning any underlying basic (biological) rationale but also overly technical discussions (e.g. lines 216 – 229) makes this text often too hard to follow for any reader beyond those specifically researching telomere dynamics in yeast (and even then...). I urge the authors to restructure their manuscript, i.e. to focus more on rationale/biology in the results section, to move all technical details to the methods section, and to discuss discordance between model and data in the discussion rather than results section.

One of the most interesting features of the paper is the inclusion of "Type B" cells, which show (non-terminal cell cycle arrest. Somewhat more introduction to this type would be interesting, particularly regarding the evidence that they may arise from a distinct yet unknown molecular origin. More importantly, a central idea behind the model is that cells can switch from type A

to type B, which was previously demonstrated to depend on the shortening of the shortest telomere. However, an important conclusion of the authors is that senescence of type B lineages independent of the (shortest) telomere length, contrasting earlier claims (and the model specification). This should be clarified. Additionally, why can't these cells switch from type B to type A? Also, in the current model, a subset of type B ("M") is introduced that cannot be discriminated from type A. Though I understand why these cannot be discriminated, it is unclear to me why this set should have a different label in a mechanistic model.

Finally, I'm rather confused by the claim – even beyond generic relevance – of the senescent cell type composition ("from a state directly linked to critically short telomeres to a state where senescence onset becomes stochastic"). Since they first demonstrate that their model works well regarding the events at work during the first days without telomerase, but provides an inferior fit during the latter days, how can the authors be certain that their claim not simply reflects that part of the model that is badly fitting. I agree that the authors make their claim based on the latter part of the well fitting part, and that there may be alternative reasons for the bad fit, but it is also possible that they are simply observing those effects that will lead to an inferior fit during later evolution.

(Remarks on code availability)

Version 2:

Reviewer comments:

Reviewer #1

(Remarks to the Author)

The authors provided exhaustive responses and discussions to my criticisms. Particularly interesting is the discussion concerning connections with carcinogenesis. This discussion also helps understanding the merits and limitations of the yeast model for some aspects of telomere dynamics.

Another interesting tidbit is the discussion of the asymmetry of telomere shortening, in response to one of my queries. Citing the response:

"This behavior is based on the asymmetry of telomere replication (Lingner et al., Science 1995; Soudet et al., 2014). Starting from a parental telomere of length L , the two newly-replicated telomeres will have a length of $L-a$ and L (a being the length of the single-stranded overhang). Thus, only one of the two newly-replicated telomeres have shortened compared to the parental one. These two telomeres are then r a n d o m l y segregated in the daughter cells."

All this is obviously correct except that I am not sure what "randomly" means here. I think the authors meant "independently". However, does an experimental proof exists that establishes this type of independence? I never heard of one. Maybe the authors might provide some evidence?

(Remarks on code availability)

I expressed a positive opinion in the initial round of review.

Reviewer #2

(Remarks to the Author)

The authors have addressed adequately all my concerns by revising the manuscript or explaining why they prefer not to. In my opinion, the revised manuscript has improved in clarity and is particularly more accessible now to the non-mathematician readers. I have no additional comments - it is ready for publication.

(Remarks on code availability)

Reviewer #3

(Remarks to the Author)

The revised version of the manuscript "Individual cell fate versus population dynamics revealed by a mathematical model linking telomeres to senescence" provides a major improvement compared to the initially submitted version. Particularly writing style, general structure and basic introduction of relevant yet specialized topics have been improved, making the manuscript far more easier to understand. Similarly, additional biological rationale is provided, making the choice of specific options far less ad hoc than in the original version of the manuscript. However, two other main concerns with the previous version have not been appropriately addressed.

While I still strongly agree that a modelling approach is often essential to gain insight into the underlying biology, the question remains what biology we're actually considering. Indeed, it remains unclear whether the specific model and derived conclusions have any relevance beyond *S. cerevisiae*. For example, the model heavily depends on the presence of type A and type B cells, but to which extent do these cell types bear relevance beyond yeast? Though I agree with the

authors' rebuttal to my similar comment on the initial version, i.e. the results have value on their own even when they cannot be extrapolated to other species, my concern is rather that the results in the revised version are still presented as if they are generally applicable. For example, in the revised version of the abstract, the authors specifically mention *S. cerevisiae*, yet only in the context of "calibration" and "validation" of the model. Subsequently, conclusions are presented as if they are generally valid for "senescent cell populations", whereas the temporal evolution mentioned will only occur in species with type A and type B cells.

Moreover, cf. my last comment on initial version, I'm still not convinced of the accuracy of the model, and hence derived predictions. Note that I fully agree that the model fits the microfluidic data very well. Yet, any mechanistic model should not only fit the data well, but also be able to make accurate, empirically verifiable predictions. Only then one can attribute value to other predictions that are hard or impossible to experimentally verify. Here, however, the model is not able to make accurate predictions for independent data, e.g. predictions regarding population growth (Figure 3b), telomere length mode (Figure 3c) or the lifespan of cells featuring a RAD51 mutation (Figure 7e) all exhibit relevant deviations from predictions. Though the authors provide biologically plausible explanations for this lack of accuracy, and the model may indeed be accurate, in my opinion additional model validation is required before biological value can be attributed to the experimentally unverified predictions made by the model, i.e. the main conclusion regarding the temporal evolution of (specifically *S. cerevisiae*) senescent cell populations.

(Remarks on code availability)

Version 3:

Reviewer comments:

Reviewer #3

(Remarks to the Author)

I had two major remaining concerns regarding this manuscript, one relating to the generic statements made by the authors regarding relevance beyond (telomerase negative) *S. cerevisiae* and one with respect to the potential of their mathematical model to make new predictions.

Though the authors claim to have addressed the former comment, their modifications are very limited. They have modified the introduction (and remainder of the manuscript) by dropping the term "in budding yeast" a couple of times, and now mention that the existence of B cells remains to be established in other species at the end of their discussion section. However, the title and abstract remained identical, and portray the results as if they are generically relevant, using the budding yeast data solely for "quantitative calibration and validation". Similarly, they claim (end of the discussion section) that "Our model also provides a solid conceptual framework for the notion ... may underlie various human diseases...". I utterly disagree on how this model could provide a solid conceptual framework to study human disease if the existence of a key component (existence of type B cells) has not yet been demonstrated beyond *S. cerevisiae*.

Regarding the demonstrated value of their mathematical model, provided data was a good start, but not proof of validity. However, the authors have added independent experimental data, validating the biological explanations provided for the "lack of fit" of their model for previous results. Particularly supplementary Figure 3C provides convincing results, demonstrating that exclusion of telomerase-independent survivors explains observed discrepancies, as predicted.

(Remarks on code availability)

Reviewer #1 (Remarks to the Author):

The paper is devoted to a comprehensive study of telomere dynamics in yeast cells, which is based on experimental data from cell cultures with daily passages and cell lineage statistics obtained from microfluidics. The idea is that combining these two types of data, it is possible to uncover detailed information about temporal trends in aging yeast cells, with possible implications for senescence of human cells, and with specific reference to carcinogenesis.

The simulation model is a complex multitype branching process with the cell types identified as being (i) the lengths of telomeres in individual yeast cells, and (ii) the cell types A and B, with the A cells inevitably dying after becoming senescent, and B cells capable of returning from senescence (with appropriate transition rules as in Figure 1, and rules related to cell cycle lengths). An additional M-type of cells seem to be B cells, indistinguishable experimentally from the A cells (see further on).

One of the most interesting phenomena observed in the experiments is temporarily reversible senescence in B-type cells, which occurs despite the continued shortening of telomeres. Many, although not all (see Figure 3 B and C), characteristics of experimental data are replicated, and some characteristics which are unobservable experimentally are strikingly illustrated by the simulations (Figure 5).

All in all, this may be the most comprehensive and realistic mathematical model of telomerase-negative cell senescence in the literature. In my opinion it is of sufficient quality to be published in the Journal, provided certain questions are addressed in a revision.

We thank the reviewer for the positive feedbacks and constructive comments.

Detailed comments:

1. I think it will be useful if the authors discuss why the A cells cannot simply be a variety of B cells with 0 returns from senescence. There may be a biological reason that I do not see, though.

We previously showed that the non-terminal arrests observed in type B cells in microfluidics were Pol32-dependent (Xu et al. 2015) and adaptation-dependent (i.e. Cdc5- and Tid1/Rdh54-dependent; Coutelier et al. 2018). This led to the interpretation that non-terminal arrested cells experienced break-induced replication repair events and adaptation to DNA damage.

In a previous statistical analysis (Martin et al. 2021), we found that the probability law governing the appearance of non-terminal arrest was distinct from the one describing senescence onset. For instance, the first non-terminal arrests in type B appear significantly earlier than the terminal arrests in type A. We thus consider that there are genetic and statistical evidence indicating that type A cells, as a whole, are not simply a subset of type B cells. We now more explicitly state these two kinds of evidence supporting distinct type A and B cells, with reference to previous works (p. 3 lines 75-79 and then p. 6 lines 168-172).

That being said, if we consider a single instance of an individual trajectory observed in a microfluidics experiment classified as type A, we cannot exclude that the terminal arrests are actually type B non-terminal arrests that failed to return to normal cell cycles. This is why we introduced the misclassified type “M” in this work

(see p. 6 lines 172-176), which can only be detected in simulations where we know the ground truth.

2. Line 83. It is quite interesting how the "telomere decay" genomic instability fits into a wider picture of genome duplication followed by chromosome mis-segregation (see papers by Watkins et al. in Nature or Dinh et al. in biorxiv <https://doi.org/10.1101/2024.04.03.587939doi>). A connection certainly exists (see eg. Lo et al. Neoplasia 4.6 (2002): 531-538 and papers citing this one). Can any details be elucidated by this model?

We thank the reviewer for these comments and suggestion we now refer in the discussion p. 12 lines 444-446. The cited papers are excellent examples of modeling cell growth in the context of evolving heterogeneity, as in our case. These examples focus on tumor growth amid increasing genomic instability. The resulting simulations have been compared to the highly complex genotypes observed in cancers, which exhibit variable gains and losses of cancer drivers, such as tumor suppressors and oncogenes. The relationship between our model and the cited examples is twofold.

On the one hand, senescent cells have been found in tumor microenvironment that contribute to senescence-associated secretory phenotype (SASP), which involves the secretion of pro-inflammatory cytokines, growth factors, proteases, and other molecules that can promote tumor growth, invasion, angiogenesis, and immune evasion. The origins of these senescent cells may be diverse, and remain to be clearly established. Whether these senescent cells derive from insufficient telomere lengthening by telomerase has not been excluded, since we know that telomeres in tumors can be often very short, despite telomerase activity (*i. e.* Xu and Blackburn, *Mol Cell* 2007 Vol. 28 Issue 2 Pages 315-27). In budding yeast, post-senescent survivors' populations have been shown to contain senescent cells, which undergo telomere shortening (Misino et al., *NAR* 2022 Vol. 50 Issue 22 Pages 12829-12843). We thus may speculate that telomere shortening and subsequent replicative senescence could contribute to cells' fitness in tumors. In this context, including our model of telomere length dynamics and replicative senescence in tumor growth model could certainly help testing this hypothesis.

On the other hand, telomerase inactivation has been shown to promote genomic instability, namely gross chromosomal rearrangements, both in humans and *S. cerevisiae* (we reviewed in Coutelier and Z. Xu, *Curr Genet* 2019 Vol. 65 Issue 3 Pages 711-716; Henninger and Teixeira, *Curr op gen & dev* 2020 Vol. 60 Pages 99-106; we can also cite Maciejowski and de Lange, *Nat rev. Mol cell biol* 2017 Vol. 18 Issue 3 Pages 175-186 and of course Lo et al., *Neoplasia* 2002). This genomic instability most certainly contributes to cells' fitness as assessed in budding yeast (see for example Beach et al., *Cell* 2017 Vol. 169 Issue 2 Pages 229-242). Knowledge on the impact of individual mutations on cells' fitness could also be extracted from genome wide screens. Thus, fitness alterations due to genomic instability could, in principle, be included in a future model. However, we are still missing a realistic mechanistic model based on experimental evidence to do so. Current experimental work in our labs aims to determine the mechanisms that initiate genomic instability during replicative senescence, and we are starting to have insights into this matter. We will therefore definitively leverage this knowledge in our future work.

3. Line 263. "Type B gradually replace (replaces?) Type A in senescent populations". I would risk saying that this observation very simply follows from a mathematical representation of the fates of the B cells as a series of failures the number of which is described by a roughly geometric distribution (thus with mean count of failures > 1), while the A cells live only through 0 failures.

Throughout the paper, "senescent" cells refer to cells in a terminally arrested state and thus the non-terminal arrests typical of type B are not included in the "senescent" population. The observation that within the senescent cells, type B cells replace type A cells over time, rather reflects the fact that type A cells are exhausted faster and that type B lineages are both longer in number of generations and average cell cycle durations (to which non-terminal arrests contribute).

4. Line 277. The M cells. Are they B misclassified as A, or simply A and B are the same cells with different random count of returns from senescence. And, could you explain what "misclassification" means, using the example of a specific cell observation?

As explained in response to the first comment, we argue that type A and type B (and their respective terminal/non-terminal arrests) are biologically distinct, based on genetic and statistical evidence. The M cells are thus fundamentally type B cells, which accidentally died during their first sequence of "non-terminal" arrests. When we single out a type A lineage trajectory from an actual experiment, we cannot definitely exclude that it is not a type B that died during non-terminal arrests, i.e. a type M. The descriptive labels on the experimental lineages can only be type A or B based on the observed behavior, but in simulations where the true type of each cell is known, we can end up with type B cells dying during their first sequence of non-terminal arrests, thus behaving like type A cells using the lineage representation. We added a more detailed explanation on lines 158-176.

5. Lines 398-399. Might you elaborate on these differences?

Sholes and colleagues (Sholes et al. 2021) used long-read Nanopore sequencing to show that the length distribution of some chromosome-end-specific telomeres is different from the others, suggesting the contribution of cis elements to the setting of the equilibrium length. O'Donnell and colleagues showed that these chromosome-end-specific differences are present in natural yeast strains and some are conserved across strains. This layer of additional refinement in the biology of telomere homeostasis supports our cautious warning that a single number (ie 27 bp in our case) for a critically short telomere should not be taken at face value. We have now expanded the text to explain these differences found in these two papers (p. 11 lines 383-388). We also include the citation of the recent preprint from Wellinger lab in which a sub-telomere -specific mechanism is proposed to explain these chromosome end-specific homeostasis.

6. Line 422-424. "We can also speculate ... is more often triggered by the telomeres that were initially ranked as second, third, fourth, etc. shortest, etc." You cite some figures in support, but maybe a more detailed discussion of this interesting hypothesis is in place? What is the model-based evidence for why and how would this happen?

This behavior is based on the asymmetry of telomere replication (*Lingner et al., Science 1995; Soudet et al., 2014*). Starting from a parental telomere of length L , the two newly-replicated telomeres will have a length of $L-a$ and L (a being the length of the single-stranded overhang). Thus, only one of the two newly-replicated telomeres have shortened compared to the parental one. These two telomeres are then randomly segregated in the daughter cells. Starting with an initial distribution of telomere length in the ancestor telomerase-negative cell, some lineages can preserve a non-shortened shortest telomere for several divisions until the initially ranked 2nd shortest telomere overtakes the shortest one to become the “new” shortest one. Because cells in which the shortest telomere reaches the critical threshold fastest are counter-selected, we can imagine that in population, we select for subpopulations in which such behavior occurs (ie the shortest telomere(s) is/are often maintained). We described this interesting phenomenon in a previous work (*Bourgeron et al. 2017*) and we have now explicitly explained the mechanisms behind this idea in the text (p. 11 lines 417-419).

Minor issue

Line 32. "how factors spuriously influencing cell viability". What does "spuriously" mean in this context?

We misused the word “spuriously” and actually meant “independently of telomeres”. The sentence has now been corrected.

Reviewer #1 (Remarks on code availability):

More specifically, I reviewed the code annotation, which seems to be sufficiently transparent. The link provided works after it is pasted into my Mozilla browser.

Reviewer #2 (Remarks to the Author):

The reviewed manuscript by Rat et al, describes a novel and original approach for mathematical modeling of growth and senescence in a population of yeast cells undergoing telomere shortening upon the depletion of telomerase. Cells in the culture are experiencing replicative senescence, non-terminal growth arrest and accidental telomere length-independent senescence. The model takes into account these processes and describes the growth and telomere dynamics of individual cells and lineages, as well as the growth on the population level, based on experimental data from single-cell lineages in microfluidics experiments. Mathematical simulations of individual cell lineages show almost identical results to the microfluidics experiments, validating the mathematical model. On the population level the model fits well the experimental data for the first 3 days upon depletion of telomerase, while later on the experimental data (but not the simulated) shows the recovery of the population, possibly due to selection of a subset of cells that bypass senescence. The authors used the mathematical model to test the contribution of the shortest telomeres, the mode and heterogeneity of telomere length, and morbidity unrelated to telomere length, to the lifespan of cell lineages and population dynamics. These experiments highlight the (under-appreciated) contribution of telomere length heterogeneity, founder effect of short telomeres, and telomere length-independent morbidity to senescence. I find these results extremely interesting and important to our understanding of population dynamics and senescence, not only in unicellular organisms such as yeast, but also in human somatic tissues, which are mostly telomerase negative. Such a model applied to human cells can help understanding the effects of compromised telomerase activity in telomere-related diseases, cancer, and clinical conditions that demand increased cell proliferation (e.g., bone marrow transplantation). It can also elucidate the contribution to senescence of increased telomere length heterogeneity and the combination of short telomeres and telomere length-independent morbidity, as observed in various clinical conditions. While beyond the scope of this manuscript, the mathematical approach reported here will hopefully provide the basis for such more complicated simulations of human cells in the future. The manuscript is clearly written and the conclusions and biological implications of the mathematical model are sound and well articulated.

We thank the reviewer for the positive feedbacks and constructive comments.

Comment:

The main issue that puzzles me is whether recombinational repair that elongates the telomeres can explain (1) the emergence of survivals and population recovery in the experimental data, and whether this can also be modeled; and (2) the behavior of type B cells, which undergo terminal senescence in a presumed telomere-length independent manner. If recombinational repair is initiated in a temporary manner upon the non-terminal arrest of type B cells, perhaps telomeres are already elongated at this stage. Then the repair pathway is silenced and telomeres resume shortening till reaching senescence at the same telomere length as type A cells. These are of course speculations, but I'd be happy if the authors elaborate on this issue a bit more. Or otherwise, what is the genetic-molecular difference between type A and type B cells? See also below.

We thank the reviewer for this very interesting question and comment. Regarding point (1), because the frequency of telomerase-independent survivors has been estimated at around \$10^{-5}\$ per division, the microfluidics experiments tracking 20-70 cell divisions per lineage, for a total of ~5000 divisions, could not capture such low frequency survivor events.

Regarding point (2), like the reviewer, we also speculate that type B cells could be precursors of survivors (we formulated this idea in *Xu et al. 2015*). The non-terminal arrests being, at least partially, Pol32, Cdc5 and Tid1-dependent, supports this hypothesis (*Xu et al. 2015, Coutelier et al., 2018*). At this stage, however, we have no evidence that telomere elongation events occurred in the non-terminal arrests of type B cells and could not model this possibility. What we can say, though, is that the model in its current state fits very well the individual type B trajectories (Fig. 1c). In general, we cannot exclude that additional mechanisms (e.g. telomere elongation during non-terminal arrests) might change some properties we uncover, but this does not invalidate the results based on the current model. As for any model, there are intrinsic limitations related to how it is built. We now mention some of these limitations in the discussion (lines 406-409).

As for the differences between *type A* and *type B*, we previously showed that the non-terminal arrests observed in type B cells in microfluidics were Pol32-dependent (*Xu et al. 2015*) and adaptation-dependent (i.e. Cdc5- and Tid1/Rdh54-dependent; *Coutelier et al. 2018*). This led to the interpretation that non-terminal arrested cells experienced break-induced replication repair events and adaptation to DNA damage. In a previous statistical analysis (*Martin et al. 2021*), we found that the probability law governing the appearance of non-terminal arrest was distinct from the one describing senescence onset. We thus consider that there are genetic and statistical evidence indicating that *type A* cells, as a whole, are not simply a subset of *type B* cells. We now more explicitly state these two kinds of evidence supporting distinct *type A* and *B* cells, with reference to our previous works (p. 3 lines 75-79 and p.6 l.168-176).

Specific comments:

1. Lines 192, 197: *There is no figure 1G.*

We apologize for misnumbering the figures. We meant “Fig. 2a”. This is now corrected.

2. Lines 241-243: *This sentence can be simplified, for example: “A difference between the experiment and the simulation that may explain this discrepancy is...”* On line 251, I suggest “Another possible explanation for the difference...”, and on line 256: “these two possible bias sources...”

We thank the reviewer for these suggestions. We have changed the text accordingly.

3. Lines 242 and 244: $\sim 3 \cdot 10^5$ and $\sim 2 \cdot 10^{-5}$, do the authors mean 3×10^5 and $\sim 2 \times 10^{-5}$? May be clearer to the broad readership.

We have replaced all instances of "." with "x" throughout the entire manuscript.

4. Lines 243-244: *Have the authors tried to add to the simulation also the emergence of post-senescence survival at the indicated rate measured experimentally ($\sim 2 \cdot 10^{-5}$)? Perhaps it can decrease the discrepancy? Or, compare experimental versus simulated population growth (as in Fig. 3B,C) for the strain lacking RAD51 (shown in Fig. 7E)?*

As mentioned above, we are currently working on modelling telomerase-independent survivors, in our follow-up project. This involves testing and adding new rules on how telomeres are elongated by homology-based mechanisms. For example, we currently lack the knowledge on which templates are used by short telomeres: are they also short telomeres or rather long ones? We do not know whether only the shortest telomeres are re-elongated or all of them, and with which probability. At the moment, we lack precise experimental data to tease apart these possible molecular mechanisms and to implement survivor emergence. We also lack single-cell data on the cell cycle duration of type I, type II and hybrid type survivors. We added a few lines to comment these points (p. 11 lines 406-409).

Allowing cell regrowth at 3-4 days (or maybe before) at a macroscopic level without modelling the molecular mechanisms would definitely reduce the discrepancy but would not provide much information. The *rad51* Δ mutant suffers from several caveats if we wanted to use to test the hypothesis that the discrepancy between experiment and simulation at later stage is due to survivor emergence. First, as we noted, *rad51* Δ display an intrinsically higher mortality rate, irrespective of telomerase inactivation. Second, the *rad51* Δ telomerase-negative strain shows an “accelerated” senescence in liquid assay. And finally, deletion of *RAD51* does not prevent completely survivor emergence (*Moore et al., Genetics 1999 Vol. 152 Issue 1 Pages 143-52*). For all these reasons, using the *rad51* Δ strain to better understand the later part of the growth curve where the discrepancy lies would not be informative. We can also add that other mutants, such as *rad52* Δ , suffer from the same limitations. Intrinsic cell mortality in *rad52* Δ is even more pronounced than in *rad51* Δ .

Overall, we believe that implementing survivor emergence is the next exciting part of our modelling endeavor, but will require much work both in terms of modelling and experimental data acquisition, and constitutes a project on its own.

5. Lines 319-323 and Figs. 6A and S5A: Note that at days 4-6, when the effects of shifting the telomere length on HSL are apparent, there is a large discrepancy between the simulated and experimental results. Please address the relevance of the simulation here.

In Fig. 3b, we indeed see discrepancy regarding cell growth appearing from day 5 onward. Still, as we discuss, we think that our model is able to recapitulate the senescence process, excluding survivor emergence. Starting from this basis, the model is internally consistent, which is why we perform most of our simulations until day 7 or 8. The HSL plot follows the same rationale: using the same model, i.e. the same rules, how is cell proliferation in bulk modified if we alter the initial telomere length distribution? In other words, while we acknowledge that the model is not perfect, we explore the behavior and the consequences of the mechanisms implemented in the model, in a self-consistent manner, which does not require a constant back-and-forth comparison with the experimental data, especially when no experimental data is available.

In any case, in Fig. 6a, we are modifying the initial telomere length distribution artificially, in a manner that cannot be done experimentally. So here, we are more in the realm of “thought experiments”, backed up by mathematical modelling and simulations.

Regarding the HSL metric specifically, we could also have chosen the point in time when growth reaches 90% of saturation, which in Fig. 3b would fall at a point where the discrepancy has not yet appeared. The simulation would then still lead to similar results (thanks to the internal consistency of the model). But the HSL is much easier to apprehend for the reader.

6. Lines 358-362: *The authors performed simulation of a strain lacking RAD51 with increased constant mortality, and observed a discrepancy between experimental and simulation (Fig. 7E). Could this discrepancy be explained by the role of RAD51 in recombinational repair and the generation of type 1 post-senescence survivors? Please clarify better the last two sentences of this paragraph.*

In Fig. 7e, the simulations are based on an increased mortality rate, independent of telomeres and telomerase. Our goal was to assess how much of the apparent senescence “acceleration” observed for *rad51Δ* could be explained simply by such telomere-independent mortality rate. The results suggest that the acceleration is partially explained by the increased telomere-independent mortality and thus partially by telomere-specific functions of Rad51.

When we introduced Rad51, we wrote: “Rad51 also binds to short telomeres and is involved in the emergence of post-senescence survivors, and perhaps other mechanisms operating at short telomeres”. These telomere-related roles could explain the remaining differences between our simulations and the experiment. As suggested by the reviewer, we now spell this out explicitly at the end of the paragraph (p. 10 lines 344-345) and also in the discussion section (p. 12 lines 437-438).

7. *Type B cells are simulated to enter senescence at a telomere length-independent manner, and also accumulate senescing cells without a telomere at all ($l=0$). I wonder if this result together with the simulation of the *rad51* mutant are not suggesting that type B cells can elongate their telomeres at the non-terminal senescence state (lines 412-415) by recombinational repair and therefore enter terminal senescence at telomere lengths that are actually longer and less heterogenous than the short and heterogenous telomeres simulated without taking into account recombinational repair.*

As we explained in the response to the main comment, we also adhere to the idea that repair events (which might be Pol32-dependent break-induced replication) at non-terminal arrests could re-elongate telomeres and even promote survivors (as we speculated in *Xu et al. 2015*). However, we have no actual data that would allow us to model this possibility and the model as it stands is able to recapitulate *type B* trajectories very well (Fig. 1c).

We now discuss this particular point if one day, we are able to refine the model to include potential elongation events, with experimental data to support them p. 11 l. 406-409.

See also the response to the reviewer’s main comment.

8. Lines 387-389: *If experimental measurement by long-read sequencing identified 70-75bp as the length of critically short telomeres, does it make sense to use this value to correct the mathematical model?*

Sholes and colleagues proposed 70-75 bp as a typical length for the critically short telomere. However, in their experiments, there was no way to assign this value to *type A* or *type B* senescence. In addition, the experiments were performed in bulk, leading to a population of heterogeneous senescent cells representing a mixture of different types with different histories. In our work, *type A* senescence is triggered at a near deterministic threshold of 27 bp, but the length of the critically short telomere of *type B* senescence is much more variable (Fig. 4e) and even evolves over time (Fig. 4d). For example, our simulations suggest that in the first 3 days, this *type-B*-specific critical length is around 50-150 bp. Overall, if we take both *type A* and *type B*, our results may not be that far from the 70-75 bp value proposed by Sholes and colleagues. Still, in our discussion, we already warn against using a single threshold length at face value. See also the response to reviewer 1's comment #5. In addition, because of the structure of the population (*type A/B* and evolution over time) and the whole distribution of critical lengths (Fig. 4e), it would be difficult to arbitrarily decide to offset the signalling length by 70-75 bp.

Finally, in another work using long-read sequencing, Kockler and colleagues propose that at lengths around 70-90 bp, telomeres could undergo recombination and senescence would be triggered for lengths below this range if recombination fails. The authors also write that these values might vary depending on the strain background and are not yet very precisely established. Taken these two studies together, as well as ours, we are still not yet in a position to precisely report an exact value for the critical length, which is likely to encompass a whole range.

9. *Figure 1C: Can the authors enlarge the examples shown for single lineages?*

We have included these as Supplementary Fig. 1a.

10. *Figure 3A: Difficult to see the extremum and 95% values. Perhaps they are not important here and can be removed from this figure?*

The reviewer is correct: the extremum and 95% values do not vary significantly in these simulations, making the curves difficult to distinguish. However, for the sake of consistency with the other graphs and to comply with the journal's requirements, we prefer to keep these values in the graph.

11. *Figure 5A: by Shortest, do you mean 'Shortest shortest'?*

The reviewer is correct. We have now changed "Shortest" into "Shortest shortest".

12. *Figure 5B-D: Perhaps it would help the reader to reverse the order of the bins listed on the right (i.e., 10th on top and 1st on the bottom).*

Thank you for this suggestion. We changed this.

13. *Figure S3B: What are the units Mo?*

"Mo" means MegaOctets. We have now added the definition in the legends.

Reviewer #3 (Remarks to the Author):

In their manuscript, Rat et al. present a mathematical model for population dynamics in telomerase-negative yeast based on replicative senescence driven by telomere shortening. The model integrates the research group's previously generated insights, is calibrated on existing experimental data, and yields several findings regarding e.g. the impact of the shortest telomere. I am convinced that this type of mathematical modelling is essential to provide mechanistic insight into how (critical) telomere shortening at a cellular level impacts the larger scale, yet I have several major concerns regarding this study and its presentation.

A first major concern is that I am currently not convinced about the relevance of the model (and hence the authors' results). Contrasting the title making a generic claim, this study focuses solely on the population dynamics of telomerase-negative yeast. Even if the model would successfully capture these dynamics, it is unclear to which extent findings can be biologically extrapolated to any other setting, even telomerase-positive yeast for that matter. Though I can imagine that these population dynamics partially reflect the population dynamics of cells in tissues of metazoans affected by critical telomere shortening (such as humans), there are simply too many differences to consider the first an accurate model for the latter. For example, the role of (telomerase positive) stem cells/progenitor cells, the role of sampling when culturing yeast, the impact of contact inhibition in metazoans, the question whether "type B" is also relevant beyond yeast etc. Note that I fully agree that yeast is a valid model system for telomere length biology and dynamics at the cellular level, but that I oppose to claims of generic relevance (as suggested by the title) at higher levels. I therefore also suggest to clarify the exact aims of this research.

While we agree that our modelling work cannot be immediately applied to human tissues, we believe that similar approaches could be conceived for human cells and that our work illustrates the benefits of modelling and simulations to reach non-trivial conclusions about telomere shortening dynamics and evolving telomerase-negative population structure, as was pointed by the other two reviewers. To clarify the aims and the model organism we consider, we have now added a sentence in the introduction (p. 4 lines 93-95).

Additionally, the work is currently presented in a very opaque way. I am well aware of the fact that presenting novel, complex models is not straightforward, as it requires both a mathematical description and a biological rationale for the description. Typically, in the results section the model and its rationale are introduced, and the methods section explains the implementation, parameter estimation, computational workarounds etc. In this manuscript, however, the results section continues to elaborate on how the model was fit to data, and why there are certain discrepancies with experimental data, with only minimal attention to what the assumptions or results imply from a biological point of view. E.g. the results section states that they started the simulations with telomere lengths randomly drawn from the distribution described in 12, 17 (lines 157/158), a few lines further (163/164): "the length of the shortest telomere in cells is computed after running the telomere shortening model described in 19" etc. Solely providing references without mentioning any underlying basic (biological) rationale but also overly technical discussions (e.g. lines 216 – 229) makes this text often too hard to follow for any reader beyond those specifically researching telomere dynamics in yeast (and even then...). I urge the authors to restructure their manuscript, i.e. to focus more on rationale/biology in the results section, to move all technical details to the methods section, and to discuss discordance between model and data in the discussion rather than results section.

We understand this comment on the organization of the paper. We have postponed technical details in Methods on the one hand – especially the ones which are cited - and gave more biological ground to the assumptions on the other hand – which allows us to better explain the *type A*, *type B* and *type M*. We have however kept the interpretation of the results in the results section rather than on the discussion section, since these interpretations have led us to new simulations and results, and seem to us necessary to comprehend our approach step by step.

One of the most interesting features of the paper is the inclusion of “Type B” cells, which show (non-terminal cell cycle arrest. Somewhat more introduction to this type would be interesting, particularly regarding the evidence that they may arise from a distinct yet unknown molecular origin. More importantly, a central idea behind the model is that cells can switch from type A to type B, which was previously demonstrated to depend on the shortening of the shortest telomere. However, an important conclusion of the authors is that senescence of type B lineages independent of the (shortest) telomere length, contrasting earlier claims (and the model specification). This should be clarified.

This work is indeed the first where we attempt to model the senescence of *type B* cells. The description, based on mathematical modelling, of how telomeres behave in *type B* cells is therefore new. In particular, the properties of the shortest telomere in *type B* cells entering senescence are unexpected but do not contradict our previous works, since they only considered *type A* senescence. We have added a more detailed description on lines 75-79 and 168-176.

Additionally, why can't these cells switch from type B to type A? Also, in the current model, a subset of type B (“M”) is introduced that cannot be discriminated from type A. Though I understand why these cannot be discriminated, it is unclear to me why this set should have a different label in a mechanistic model.

As we also explained in response to reviewers 1's and 2's comments, we previously showed that the non-terminal arrests observed in *type B* cells in microfluidics were Pol32-dependent (Xu *et al.* 2015) and adaptation-dependent (i.e. Cdc5- and Tid1/Rdh54-dependent; Coutelier *et al.* 2018). This led to the interpretation that non-terminal arrested cells experienced break-induced replication repair events and adaptation to DNA damage.

In a previous statistical analysis (Martin *et al.* 2021), we found that the probability law governing the appearance of non-terminal arrest was distinct from the one describing senescence onset. For instance, the first non-terminal arrests in *type B* appear significantly earlier than the terminal arrests in *type A*. Once a cell experiences a non-terminal arrest, it can therefore no longer follow a *type A* trajectory.

We thus consider that there are genetic and statistical evidence indicating that *type A* cells, as a whole, are not simply a subset of *type B* cells. We now more explicitly state these two kinds of evidence supporting distinct *type A* and *B* cells, with reference to our previous works (p. 3 lines 75-79 and p. 6 lines 168-172).

The *M* cells are fundamentally *type B* cells, which accidentally died during their first sequence of “non-terminal” arrests. When we single out a *type A* lineage trajectory

from an actual experiment, we cannot definitely exclude that it is not a *type B* that died during non-terminal arrests, i.e. a *type M*. The descriptive labels on the experimental lineages can only be *type A* or *B* based on the observed behavior, but in simulations where the true type of each cell is known, we can end up with *type B* cells dying during their first sequence of non-terminal arrests, thus behaving like *type A* cells using the lineage representation. Thus, the type M label does not change the mechanistic basis of the model; it is only informative. For example, this allows us to make sure that the proportion of experimental lineages that would have been misclassified according to our model remains low. Among all the experimental generations of senescence onset of type A lineages, only a few could instead correspond to a first non-terminal arrest in a type M lineage such that using this data to fit senescence entry in type A is relevant. We have now better explained this point (lines 172-176).

Finally, I'm rather confused by the claim – even beyond generic relevance – of the senescent cell type composition (“from a state directly linked to critically short telomeres to a state where senescence onset becomes stochastic”). Since they first demonstrate that their model works well regarding the events at work during the first days without telomerase, but provides an inferior fit during the latter days, how can the authors be certain that their claim not simply reflects that part of the model that is badly fitting. I agree that the authors make their claim based on the latter part of the well fitting part, and that there may be alternative reasons for the bad fit, but it is also possible that they are simply observing those effects that will lead to an inferior fit during later evolution.

We must first highlight that our model, calibrated on single-cell data, fits the microfluidics experiments extremely well quantitatively (Fig. 2b), which is far from trivial since multiple constrained parameters had to be optimized simultaneously. The excellent agreement even allowed us to simulate microfluidics data that are indistinguishable from experimental ones (Fig. 2c). We should also point out that the simulations at the population level (Fig. 3) are not fitted to the experiment, i.e. no parameter was adjusted to better correspond to the experimental curve, but directly compared. Therefore, we estimate that our simulations recapitulate the “senescence” part of the growth curve quite well. Since the major mechanistic difference between microfluidics data and population experiments is the absence of survivor emergence in the former (due to the low frequency of survivor emergence), we propose that the discrepancy between simulation and experiment at the population level likely stems from the telomere-independent survivors, which we did not model. Thus, our results regarding the simulated population structure are not necessarily invalidated by this discrepancy. We also discuss other reasons in the text (p. 7).

Point-by-Point Responses to Reviewers

Reviewer #1 (Remarks to the Author):

The authors provided exhaustive responses and discussions to my criticisms. Particularly interesting is the discussion concerning connections with carcinogenesis. This discussion also helps understanding the merits and limitations of the yeast model for some aspects of telomere dynamics.

Another interesting tidbit is the discussion of the asymmetry of telomere shortening, in response to one of my queries. Citing the response:

"This behavior is based on the asymmetry of telomere replication (Lingner et al., Science 1995; Soudet et al., 2014). Starting from a parental telomere of length L , the two newly-replicated telomeres will have a length of $L-a$ and L (a being the length of the single-stranded overhang). Thus, only one of the two newly-replicated telomeres have shortened compared to the parental one. These two telomeres are then randomly segregated in the daughter cells."

All this is obviously correct except that I am not sure what "randomly" means here. I think the authors meant "independently". However, does an experimental proof exists that establishes this type of independence? I never heard of one. Maybe the authors might provide some evidence?

We thank the reviewer for this comment. In fact, we mean independence in the segregation of the strands in mitosis, i.e., each daughter inherits one of the two telomeres, and we randomly choose (with equal probability $\frac{1}{2}$) which daughter inherits the shortened telomere and which the non-shortened one.

We believe the reviewer is referring to the "immortal strand hypothesis", which has been proposed for stem cell divisions. This hypothesis suggests that a stem cell can distinguish between chromatids based on their template age: the self-renewing daughter stem cell inherits the chromatids with older templates, while the differentiating daughter cell receives those with newer templates. While our aim is not to confirm or refute this intriguing theory, we would like to highlight that the DNA end replication problem, in principle, remains independent of the origin, age, or fate of the daughter strands. Each strand features both a leading and lagging end, meaning that, regardless of 'age,' each strand is subject to the shortening of one of its extremities, as described by the DNA end replication problem model. This was already pointed out by P. Lansdorp in Cell 2007 Vol. 129 Issue 7 Pages 1244-7: « It has been suggested that stem cells that show immortal strand segregation can divide more times than other cells because telomere shortening is prevented (Karpowicz et al., 2005; Potten et al., 2002). However, in theory, only the 3' end of template strands is protected against replicative shortening in cells that retain DNA template strands. This is because

the 5' end of DNA template strands must be processed following replication in order to create a single strand 3' overhang required for telomere function (Lingner et al., 1995). Thus telomere losses are predicted to occur in stem cells whether or not template strands are retained. »

Reviewer #1 (Remarks on code availability):

I expressed a positive opinion in the initial round of review.

Reviewer #2 (Remarks to the Author):

The authors have addressed adequately all my concerns by revising the manuscript or explaining why they prefer not to. In my opinion, the revised manuscript has improved in clarity and is particularly more accessible now to the non-mathematician readers. I have no additional comments - it is ready for publication.

We thank this reviewer for this positive comment.

Reviewer #3 (Remarks to the Author):

The revised version of the manuscript "Individual cell fate versus population dynamics revealed by a mathematical model linking telomeres to senescence" provides a major improvement compared to the initially submitted version. Particularly writing style, general structure and basic introduction of relevant yet specialized topics have been improved, making the manuscript far more easier to understand. Similarly, additional biological rationale is provided, making the choice of specific options far less ad hoc than in the original version of the manuscript. However, two other main concerns with the previous version have not been appropriately addressed.

*While I still strongly agree that a modelling approach is often essential to gain insight into the underlying biology, the question remains what biology we're actually considering. Indeed, it remains unclear whether the specific model and derived conclusions have any relevance beyond *S. cerevisiae*. For example, the model heavily depends on the presence of type A and type B cells, but to which extent do these cell types bear relevance beyond yeast? Though I agree with the authors' rebuttal to my similar comment on the initial version, i.e. the results have value on their own even when they cannot be extrapolated to other species, my concern is rather that the results in the revised version are still presented as if they are generally applicable. For example, in the revised version of the abstract, the authors specifically mention *S. cerevisiae*, yet only in the context of "calibration" and "validation" of the model. Subsequently, conclusions are presented as if they are generally valid for "senescent cell populations", whereas the temporal evolution mentioned will only occur in species with type A and type B cells.*

We apologize for not clearly emphasizing that our research involves data extraction and analysis specifically from yeast, with models developed accordingly. We have revised our text to better highlight the limitations of extrapolating our findings to broader contexts of replicative senescence and to reiterate that all data used in our study were obtained and analyzed specifically for budding yeast. Please refer to p. 4 (Introduction) and p. 10-12 (Discussion) for these clarifications.

*Moreover, cf. my last comment on initial version, I'm still not convinced of the accuracy of the model, and hence derived predictions. Note that I fully agree that the model fits the microfluidic data very well. Yet, any mechanistic model should not only fit the data well, but also be able to make accurate, empirically verifiable predictions. Only then one can attribute value to other predictions that are hard or impossible to experimentally verify. Here, however, the model is not able to make accurate predictions for independent data, e.g. predictions regarding population growth (Figure 3b), telomere length mode (Figure 3c) or the lifespan of cells featuring a RAD51 mutation (Figure 7e) all exhibit relevant deviations from predictions. Though the authors provide biologically plausible explanations for this lack of accuracy, and the model may indeed be accurate, in my opinion additional model validation is required before biological value can be attributed to the experimentally unverified predictions made by the model, i.e. the main conclusion regarding the temporal evolution of (specifically *S. cerevisiae*) senescent cell populations.*

We thank reviewer #3 for his positive comment on the very good agreement between microfluidic data and simulated data obtained by our model. We think that this result in itself is of high value, even without predictions on population experiments. In fact, these results are obtained on a comprehensive set of microfluidic data, retaining all lineages, taking into account a great deal of complexity while remaining sparse in term of the number of parameters fitted. In particular, the opposite behavior of *type A* vs. *type B* lineages with respect to the senescence signal (fully deterministic vs. uniformly distributed) appears to be a result of our approach.

Reviewer #3 criticizes the validity of our model, based on discrepancies on 3 figures: Fig. 3b, Fig. 3c and former Fig. 7e (now Supplementary Figure 6c). We would like to distinguish former Fig. 7e (now Supplementary Figure 6c) from the two others. Indeed, in Fig. 7e (now Supplementary Figure 6c), we compare our model, the components of which are explicitly known, to microfluidics-based data of *rad51Δ* mutant. The goal here is not to “fit” perfectly the experimental data, but rather to evaluate how much of the experimental data can be explained by the model, which includes increased basal mortality rate but no direct interaction with telomeres. The conclusion is nuanced: much, but not all, of the data is already explained by the model. We therefore show that such a modelling approach allowed us to evaluate the true contribution of a factor to telomere biology and distinguish it from an indirect effect, e.g. here through increased basal mortality.

Coming back to Fig. 3b and 3c, we agree that they correspond to the validation part of our modelling approach and should show that the model can recapitulate the experimental data. It is well established that the re-growth of the population of cells at later time points corresponds to the emergence of post-senescence survivors. Since our model was not designed to take them into account in the first place, we estimated that the divergence between experimental and simulated curves was expected. But we do understand the reviewer's concern. To better delineate that simulations based on our model can closely match the senescence part of the growth curve in Fig. 3b, we now compare simulations based on our model to data from Lydeard et al. 2007 Nature (Fig. 4b therein), where they proved that *pol32Δ* mutant cannot form post-senescence survivors.

As one can appreciate in the new supplementary Figure 3c, our simulations align well with these experimental data, indicating that, as designed, our model can predict population growth in telomerase-negative cells as long as telomerase-independent survival pathways are not considered.

While we think that in former Fig. 7e (now Supplementary Figure 6c), the deviation between simulations and experimental data should not be interpreted as a discrepancy, but rather as a way to gain insight into the nature of the contribution of Rad51 to senescence, we provide an additional comparison between simulations and growth of a *population* of *rad51Δ* cells, using new experimental data (new Supplementary Figure 6d). This analysis adds another layer of information, i.e. the contribution of Rad51 to senescence at the population level, and provides an opportunity for us to better illustrate the presence or absence of deviation between simulation and experimental data.